# GUI-Spotlight: Adaptive Iterative Focus Refinement for Enhanced GUI Visual Grounding

## Abstract

Multimodal large language models (MLLMs) have markedly expanded the competence of graphical user-interface (GUI) systems, propelling them beyond controlled simulations into complex, real-world environments across diverse platforms. However, practical usefulness is still bounded by the reliability of visual grounding, i.e., mapping textual references to exact on-screen elements. This limitation prevents the system from accurately performing pointer-level actions such as clicking or dragging. To address it, we introduce GUI-Spotlight—A model trained for *image-grounded reasoning* that dynamically invokes multiple specialized tools to iteratively narrow its focus to the relevant region of the screen, thereby substantially improving visual grounding accuracy. On the ScreenSpot-Pro benchmark, GUI-Spotlight trained with only $18.5K$ training samples achieves $52.8\%$ accuracy, surpassing V2P-7B ($50.6\%$ with $9.6M$ training samples) and GTA-1-7B ($50.1\%$ with $1.56M$ training samples).

## 1 Introduction

Recent rapid advances in multimodal (generative) vision-language pre-training—e.g., visually conditioned autoregressive decoders (Luo et al., 2022) and more recent multimodal LLMs—have driven swift progress in GUI agents (Xie et al., 2024; Yang et al., 2025b). Nevertheless, current GUI agents still lack robust, fine-grained visual grounding, making it difficult to translate *what* to do into *where* to act on complex, dynamically changing screens (Jang et al., 2024; Xie et al., 2025). As a result, they struggle to reliably perform pixel-level operations—such as precise clicking, dragging, and region selection—thereby constraining the reliability and scalability of end-to-end execution. (Cheng et al., 2024; Gou et al., 2025).

Recent studies have employed supervised fine-tuning (SFT) and reinforcement learning (RL) to train these models (Gou et al., 2025; Qin et al., 2025); however, their performance remains suboptimal on complex or high-resolution user interfaces. For example, on the high-resolution GUI visual grounding benchmark ScreenSpot-Pro (Li et al., 2025), recently released 7B models achieve only around $50\%$ accuracy (Yang et al., 2025a; Gu et al., 2025; Tang et al., 2025), which is not practical.

To overcome this limitation, we propose GUI-Spotlight, a novel GUI visual grounding model that *thinks with the image* and dynamically narrows its focus like a spotlight, iteratively homing in on the target, inspired by attention mechanisms that highlight discriminative regions. (Shu et al., 2022). To achieve this, GUI-Spotlight is equipped with a set of specialized visual tools——*crop*, *extract*, and *find color*——that allow it to iteratively interrogate sub-regions of the screen and progressively refine its search until the target is pinpointed with high precision. As shown in Fig. 1, given the user's original instruction *Click the Send button* and the original screenshot *Image 0*, GUI-Spotlight iteratively invokes tools to progressively narrow its focus to the precise click location. After each invocation, the newly cropped image is appended to the dialogue history. The model returns the final answer once coordinate confidence is sufficient.

GUI-Spotlight is trained in three stages. In Stage 1, we collect multi-turn tool-usage dialogues and warm up the model via SFT. In Stages 2 and 3, we conduct reinforcement learning with a modified Group Sequence Policy Optimization (GSPO) algorithm (Zheng et al., 2025), enabling the model to learn when and how to use tools effectively, yielding a robust policy that continually improves and substantially boosts visual grounding accuracy. Details are provided in Section 3.2.

**Original Input:** *Click Send button & Image 0*
**Action 1:** extract(Image_0, right, bottom)
**Observation 1:** *Image 1*
**Action 2:** crop(Image_1, (300, 430), (600, 500))
**Observation 2:** *Image 2*
**Answer:** answer(Image_2, (260, 30))

Figure 1: GUI-Spotlight pipeline. Orange text denotes the user's original input; blue text indicates the image provided in each dialogue turn; red text indicates the command generated by the model in that turn. Red boxes highlight the newly cropped images produced by the model's command.

Our contributions are as follows:

1. We introduce GUI-Spotlight, a think-with-image visual grounding model that performs iterative spotlighting with tool coordination; it achieves **52.8%** accuracy on SCREENSPOT-PRO and **23.4%** on UI-Vision, substantially outperforming comparable 7B baselines.

2. We modify GSPO for multi-tool reinforcement learning, yielding a stable training procedure that improves sample efficiency and final grounding accuracy.

3. We document and consolidate our attempts—including negative results—on algorithms, reward design, and training settings, providing practical insights for agentic visual grounding models with coordinated tool use.

## 2  RELATED WORK

**GUI grounding.** Recent progress in GUI agents has been propelled by specialised GUI–grounding models that map natural-language references to screen coordinates (You et al., 2024). UGround (Gou et al., 2025) scaled data collection to a universal, cross-platform backbone, while OS-Atlas (Wu et al., 2024) expanded synthetic coverage and improved out-of-distribution transfer. UI-TARS (Qin et al., 2025) explored capacity scaling with native mouse-and-keyboard action spaces, and Aguvis (Sarch et al., 2025) proposed a modular "ground-then-plan" pipeline for fully open-source autonomy. Together, these works highlight data scale, model capacity, and modular design as complementary levers, yet accuracy on dense, high-resolution, cluttered interfaces remains challenging. More broadly, prototype-based consistency regularization can stabilize localization in clutter (Xu et al., 2025; 2022).

**Reinforcement learning for grounding.** RL offers a complementary path by casting localization as sequential decision-making. UniVGR1 (Bai et al., 2025b) iteratively refines boxes and attains state-of-the-art results on RefCOCO-style corpora; GROUNDED RL FOR VISUAL REASONING ties rewards to localization correctness for multi-step reasoning (Sarch et al., 2025). In GUI automation, self-evolutionary RL reduces off-target actions without extra labels (Yuan et al., 2025). GROUND-R1 (Cao et al., 2025) further shapes rewards to balance overlap, textual relevance, and action efficiency, improving generalization to unseen categories and layouts. Despite these advances, high-resolution or heavily cluttered screens remain difficult—motivating our focus-refinement approach.

## 3  METHOD

### 3.1  AGENTIC INTERACTION FRAMEWORK

**Inference Pipeline** As shown in Algorithm 1, Given a text description $d$ and the original image $I_0$, we maintain a registry $R = \{ i \mapsto (I_i, \boldsymbol{\delta}_i) \}$, where $\boldsymbol{\delta}_i$ denotes the top-left offset of $I_i$ w.r.t. $I_0$, and a message history $\mathcal{H}$ initialized with $(d, I_0)$. At round $t$, we send $\mathcal{H}$ to the model, which returns

either $Action(i, \text{Tool}, \text{args})$ or $Stop(i, (x_{\text{rel}}, y_{\text{rel}}))$. For $Action$, executing the tool on $I_i$ yields a new image $I_{i+1}$, an information message describing the image and the offset $\boldsymbol{\delta}_{i+1}$ of image $I_{i+1}$ relative to the original image. Register $I_j, \boldsymbol{\delta}_{i+1}$ in $R$, and append the result to $\mathcal{H}$. When $Stop$ is returned, the absolute coordinate on $I_0$ is calculated and returned.

---

**Algorithm 1:** GUI-Spotlight Inference Pipeline

---

**Input:** Text description $d$ of the target element; original image $I_0$.
**Output:** Absolute coordinate of the element $(x_{\text{abs}}, y_{\text{abs}})$ on $I_0$ or None.
1 **Registry** $R = \{ i \mapsto (I_i, \boldsymbol{\delta}_i) \mid i \in \mathbb{N} \}$, where $I_i$ is the $i$-th image, $\boldsymbol{\delta}_i = (\delta_i^x, \delta_i^y)$ is the top-left offset
  w.r.t. $I_0$, and $I_0$ is the original image.
2 **Initialization**: assign $\boldsymbol{\delta}_0 = (0,0)$ to $I_0$; Message History $\mathcal{H} \leftarrow \{(d, I_0)\}$.
3 **for** $t = 1$ **to** $T_{\max}$ **do**
4     $Action \leftarrow \text{Model}(\mathcal{H})$
5     **if** $Action = \text{Stop}(i, (x_{\text{rel}}, y_{\text{rel}}))$ **then**
6         $(x_{\text{abs}}, y_{\text{abs}}) \leftarrow R[i].\boldsymbol{\delta}_i + (x_{\text{rel}}, y_{\text{rel}})$
7         **return** $(x_{\text{abs}}, y_{\text{abs}})$
8     **else if** $Action = \text{Tool}(i, \text{args})$ **then**
9         $(I_{i+1}, \text{info}, \boldsymbol{\delta}_{i+1}) \leftarrow \text{Tool}(R[i].I_i, \text{args})$
10        $R[i+1] \leftarrow (I_{i+1}, \boldsymbol{\delta}_{i+1})$
11        $\mathcal{H} \leftarrow \mathcal{H} \cup \{(I_{i+1}, \text{info})\}$
12 **return** None

---

**Tool Functions.** We design three visual grounding tools; their functionality, inputs, and outputs are summarized in Table 1. `extract`: quadrant crop by position for coarse focus narrowing. `find_color`: color-guided focusing— slides a window to locate the region of closest color match to a target RGB by minimizing the perceptual color difference ($\Delta E$, the Euclidean distance in CIE Lab space), then extracts a centered crop. `crop`: rectangular crop specified by opposite corners for fine-grained focus. All tools return the cropped image, an information message, and the top-left offset of the crop relative to the original image.

Table 1: Tool functions used in GUI-Spotlight.

| Function | Function Logic | Input | Output |
|---|---|---|---|
| `extract` | Quarter crop ($\frac{1}{2}W \times \frac{1}{2}H$) by position; validate options; enforce minimum crop size; compute top-left offset. | Image; `x_pos` $\in\{$left, center, right$\}$; `y_pos` $\in\{$top, center, bottom$\}$. | Image; Info; offset $(\Delta x, \Delta y)$ or None. |
| `find_color` | Scan $10\times10$ patches (stride 10); pick minimal $\Delta E$ (CIE Lab) to target RGB; center a $ws \times ws$ window. | Image; `target_rgb` $= (r, g, b)$. | Image; Info; offset $(\Delta x, \Delta y)$ or None. |
| `crop` | Rectangular crop with bounds/order/min-size checks; optional $\pm 1$px adjustment for edge case; compute offset. | Image; `top_left` $= (x_1, y_1)$; `bottom_right` $= (x_2, y_2)$. | Image; Info; offset $(\Delta x, \Delta y)$ or None. |

## 3.2 TRAINING

In this section, we present the datasets we collected for training, the three-stage training pipeline, and the reward formulation used during reinforcement learning.

### 3.2.1 DATASET

***Data Collection*** During our investigation, we observed that many open-source datasets already exist at low resolution. To provide more challenging tasks during training and thereby encourage deeper reasoning, we collected an additional 15K high-resolution samples. Specifically, we employ a Selenium-based headless browser to batch-load webpages at a fixed resolution and automatically detect common interactive elements. For visible elements of reasonable size, we extract their readable text, crop element-level images from full-page screenshots, and store the text together with

bounding-box metadata, organized per site. This pipeline enables large-scale collection and cleaning, resulting in a consistent dataset of clickable components for downstream training and evaluation.

***Data Cleaning*** Raw instruction-following datasets often contain significant noise, such as blurry screenshots, ambiguous instructions, or inaccurate annotations. To ensure the quality of our training data, we developed a rigorous filtering pipeline to curate a high-fidelity dataset. First, we perform image-level pre-filtering, discarding images if their Laplacian variance is below 100.0 (Clarity) or if the ground-truth bounding box covers less than 1% of the image area (Visibility).

The core of the filtering process uses the `Qwen2.5-VL-72B` model to audit each instruction-response pair via the following three evaluations:

- **Instruction Quality (IQ):** To filter out unclear instructions, the model rates clarity and uniqueness on a 0–10 scale and filters out ambiguity by accepting only those scoring $\geq 6$.
- **Bounding Box Accuracy (BA):** To verify label accuracy, the model's prediction ($B_p$) is compared against the ground-truth ($B_{gt}$). The resulting accuracy score, also on a 0-10 scale, must be $\geq 6$, as determined by the formula $S_{BA} = 5 \cdot \frac{|B_p \cap B_{gt}|}{|B_{gt}|} + 5 \cdot \frac{|B_p \cap B_{gt}|}{|B_p|}$.
- **Consistency (CON):** To ensure the model's interpretation is stable and not coincidental, this self-verification step requires the Intersection over Union (IoU) between two independently generated boxes ($B_{IQ}, B_{BA}$) to be at least 0.40, as calculated by $IoU = \frac{|B_{IQ} \cap B_{BA}|}{|B_{IQ} \cup B_{BA}|}$.

A sample is retained only if it passes all three filters. We applied this pipeline to both the public UGround dataset (Gou et al., 2025) and our newly collected high-resolution data. On the UGround dataset, the process retained approximately 50% of the data as a high-quality subset. For our high-resolution dataset, we enhanced the pipeline with additional functions to filter for websites in major languages and recognizable interfaces, yielding a refined dataset of $11.6K$ samples. This comprehensive cleaning ensures a consistent standard of quality across both datasets used in our work.

### 3.2.2 THREE STAGES TRAINING

GUI-Spotlight training was carried out in three distinct stages. The training pipeline and algorithms are presented below, and the full hyperparameter settings are provided in the Appendix A.1.

***Stage 1*** We first executed the same inference pipeline with `Qwen2.5-VL-72B` (Bai et al., 2025a) on the filtered UGround dataset and collected 2561 multi-turn dialogue trajectories with tool invocations. We then used these trajectories to warm up the initial models via supervised imitation, starting from `UI-TARS-1.5-7B` (Qin et al., 2025) and `Qwen2.5-VL-7B-Instruct` (Bai et al., 2025a). This stage teaches the models to compose multiple tools, providing a solid initialization for subsequent RL.

***Stage 2*** We further optimize the model via reinforcement learning using 12K samples from the filtered UGround dataset, and we modify the original GSPO (Zheng et al., 2025) objective as follows:

$$\mathcal{J}_{\text{Ours}}(\theta) = \mathbb{E}_{x \sim \mathcal{D}, \{y_i\}_{i=1}^G \sim \pi_{\theta_{\text{old}}}(\cdot \mid x)} \left[ \frac{1}{G} \sum_{i=1}^G \min\left( s_i(\theta)\, \widehat{A}_i,\ \text{clip}\left(s_i(\theta),\, 1-\varepsilon,\, 1+\varepsilon\right) \widehat{A}_i \right) \right] + \lambda\, \mathcal{J}'(\theta)$$

where

$$\widehat{A}_i = \frac{r(x, y_i) - \text{mean}\left(\{r(x, y_j)\}_{j=1}^G\right)}{\text{std}\left(\{r(x, y_j)\}_{j=1}^G\right)}, \quad s_i(\theta) = \exp\left( \frac{1}{|y_i|} \sum_{t=1}^{|y_i|} \log \frac{\pi_\theta(y_{i,t} \mid x, y_{i,<t})}{\pi_{\theta_{\text{old}}}(y_{i,t} \mid x, y_{i,<t})} \right).$$

$$\mathcal{J}'(\theta) = \frac{1}{\sum_{b=1}^B \sum_{t=1}^{L_b} C_b\, M_{b,t}\, +\, \varepsilon} \sum_{b=1}^B \sum_{t=1}^{L_b} C_b\, M_{b,t}\, \log \pi_\theta\left(y_{b,t} \mid s_{b,t}\right)$$

$B$ is the batch size, $L_b$ is the sequence of sample $b$, $\lambda > 0$ is a mixing weight; in stage 2 we set $\lambda = 1$; and $\mathcal{J}'(\theta)$ is computed by first filtering to the subset of samples whose outputs are both format-valid and result-correct, and then averaging the token-level cross-entropy over this subset. $M_{b,t} \in \{0, 1\}$ is the completion mask. And $C_b \in \{0, 1\}$ is the sample mask for result-correct and format-correct cases. $\varepsilon > 0$: a tiny constant to avoid division by zero.

This auxiliary term $\mathcal{J}'(\theta)$ is introduced to stabilize reinforcement learning in multi-turn tool-use scenarios. In its absence, broad autonomous exploration often leads the model to generate non-parseable tool formats, resulting in sparse and volatile rewards. Such instability induces high-variance gradients and excessively large policy updates, which in turn cause parameter drift and ultimately lead to training collapse. A detailed analysis is provided in Section 4.1.

***Stage 3*** Using 4000 high-resolution samples that we collected, we further refined training once the tool-call format had stabilized. Specifically, we reduced the mixing weight to $\lambda = 0.01$ and revised the sample mask $C_b$ in $\mathcal{J}'(\theta)$: rather than retaining all results or only format-correct cases, we applied bucketed uniform sampling across tool types.

Bucket construction:

$$S_t = \{\, b : \text{tool}(b) = t, \ \text{correct}(b) = 1, \ \text{format}(b) = 1 \,\}, n_{\min} = \min_{t \in \mathcal{T}} |S_t|, \hat{S}_t \subseteq S_t, \ |\hat{S}_t| = n_{\min}.$$

Sample mask:

$$C_b := \begin{cases} 1, & b \in \bigcup_{t \in \mathcal{T}} \hat{S}_t, \\ 0, & \text{otherwise}. \end{cases}$$

Where $t \in \mathcal{T}$ is the tool type; $b$ is the sample index. $S_t$ is bucket for tool $t$.

The evolution of test accuracy on the ScreenSpot-Pro benchmark across the three training stages is shown in Figure 2. Using `UI-TARS-1.5-7B` as an base model: Stage 1: We perform one epoch of SFT on 2561 trajectories. After warm-up, the model learns to invoke multiple tools but remains under-aligned. Stage 2: We train on 12K examples with RL, yielding a substantial accuracy gain. Stage 3: we then introduce additional 4000 samples to encourage exploration, leading to a further improvement in accuracy.

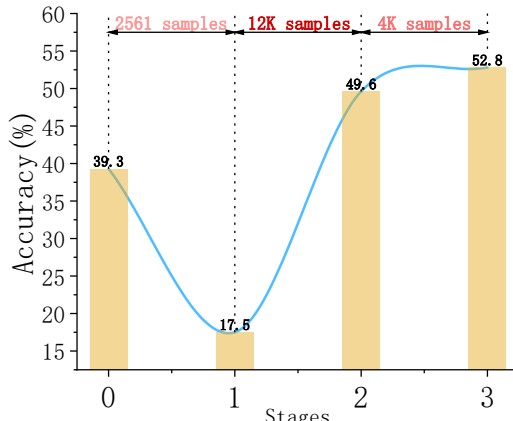

Figure 2: ScreenSpot-Pro accuracy over training.

### 3.2.3 REWARD DESIGN

We combine five rewards into a weighted sum

$$R = \sum_{k=1}^{5} \alpha_k r_k, \ (\alpha_1, \ldots, \alpha_5) = (0.30, 0.25, 0.05, 0.20, 0.20), \ \sum_k \alpha_k = 1.$$

They are summarized in Table 2. $r_1$ `Answer` is a sparse reward for a correct final answer. $r_2$ `Crop` uses IoU to provide dense reward toward the ground-truth region. $r_3$ `Extract` and $r_4$ `Find_Color` supply binary intermediate feedback for quadrant/color-guided focusing. $r_5$ `Format` checks the syntactic validity of tool calls to improve the stability of training. The exploration of how

Table 2: Reward components used for RL training. $B^\star$ denotes the ground-truth bounding box.

| # | Name | Definition | Type | Weight |
|---|------|-----------|------|--------|
| $r_1$ | Answer | 1 if the final answer the predicted coordination $(\hat{x}, \hat{y})$ lies inside the $B^\star$ $(x_1, y_1, x_2, y_2)$; 0 otherwise. | *sparse* | 0.3 |
| $r_2$ | Crop | For each `<crop>` call $i$ we compute $\text{IoU}_i = \frac{|\hat{B}_i \cap B^\star|}{|\hat{B}_i \cup B^\star|} \in [0, 1]$. | *dense* | 0.25 |
| $r_3$ | Extract | Each quadrant extracted by `<extract>` yields 1 if it fully contains $B^\star$, else 0. The reward is their mean. | *sparse* | 0.05 |
| $r_4$ | Find_Color | Returns 1 when the $200 \times 200$ color-match window covers $B^\star$; 0 otherwise. | *sparse* | 0.2 |
| $r_5$ | Format | 1 if the assistant's tool call is syntactically valid; 0 otherwise. | *sparse* | 0.2 |

different reward designs affect the final model performance are shown in Section 4.2.

## 4 EMPIRICAL INSIGHTS

Throughout this study, we systematically evaluated how different algorithms and reward designs affect the model's final performance; the experimental records are presented below.

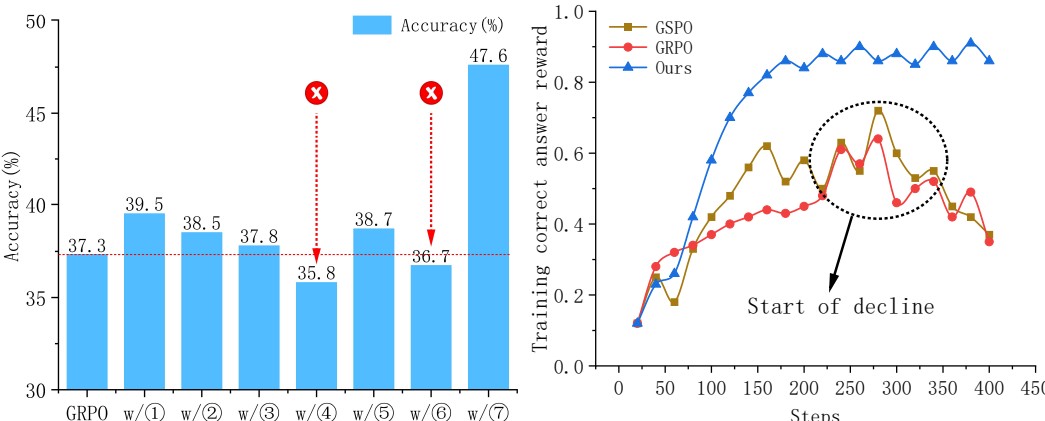

Figure 3: **Left**: Impact of different RL variants. **Right**: A comparison of algorithm training dynamics . ❌ denotes *discarded*. Items ①–⑦ are described in the first paragraph of Section 4.1.

## 4.1 RL ALGORITHM SELECTION

To investigate how different RL algorithms perform on the multi-turn, tool-using GUI visual grounding task, we benchmark a suite of GRPO-based (Shao et al., 2024) improvements alongside our own variants. Specifically, the evaluated techniques include ① sequence-level importance-ratio sampling (Zheng et al., 2025), ② Clip-Higher (Yu et al., 2025), ③ KL-term removal (Yu et al., 2025), ④ retaining only the top $p\%$ most-uncertain prompts (Wang et al., 2025), and ⑤ adding a positive-example LM loss (Yue et al., 2025). In addition, we introduce two of our own designs: ⑥ continuously updating the reference policy, and ⑦ tool-filtered positives with an additional cross-entropy loss. To isolate the effect of the RL objective, we first conduct a Stage-1 warm-up and then compare RL algorithms under identical settings. Concretely, we initialize from the same SFT checkpoint of UI-TARS-1.5 trained for one epoch on 2561 multi-turn tool-invocation trajectories, run 400 RL steps for each method, and evaluate on the SCREENSPOT-PRO benchmark; training parameters are provided in the Appendix A.2. As shown in the left panel of Figure 3. In our setting, selecting only the highest $p\%$ most uncertain prompts and continuously updating the reference policy both degrade accuracy. Therefore, we discard these two modifications and keep the remaining improvements.

Moreover, as shown in the right panel of Figure 3, we observe that the tool-filtered positives with an additional cross-entropy loss effectively prevents RL collapse on this task. Vanilla GRPO or GSPO begins to oscillate around 300 steps, with outputs increasingly violating the tool-call syntax, leading to a gradual drop in accuracy. In contrast, once we add the additional cross-entropy loss , the training curve no longer degrades and instead continues to improve.

## 4.2 REWARD DESIGN

To investigate how different reward formulations affect model performance on the multi-turn, tool-using GUI visual grounding task, we conducted experiments using the same settings as Section 4.1.

First, We study how the different types of Answer reward affects the final performance. Specifically, we compare two formulations:

1. **Binary sparse reward**: assign $r_{\text{answer}} = 1$ if the predicted click $(x, y)$ lies inside the ground-truth box $(x_1, y_1, x_2, y_2)$; otherwise $r_{\text{answer}} = 0$.
2. **Center-shaped dense reward**: if $(x, y)$ is inside the box, let $c_x = (x_1 + x_2)/2$ and $c_y = (y_1 + y_2)/2$ be the box center, and $(x_2 - x_1)/2$, $(y_2 - y_1)/2$ the half-width/half-height. Define the normalized Chebyshev distance

$$d = \max\left(\frac{|x - c_x|}{(x_2 - x_1)/2}, \frac{|y - c_y|}{(y_2 - y_1)/2}\right) \in [0, 1], \quad \text{closeness} = 1 - d,$$

and set

$$r_{\text{answer}} = 1 + \text{bonus}_{\text{max}} \cdot (\text{closeness})^\gamma \text{(if inside)}, \quad r_{\text{answer}} = 0 \quad \text{(otherwise)},$$

where $\gamma \geq 1$ shapes the curvature around the center and $\text{bonus}_{\text{max}}$ controls the maximum extra credit at the center.

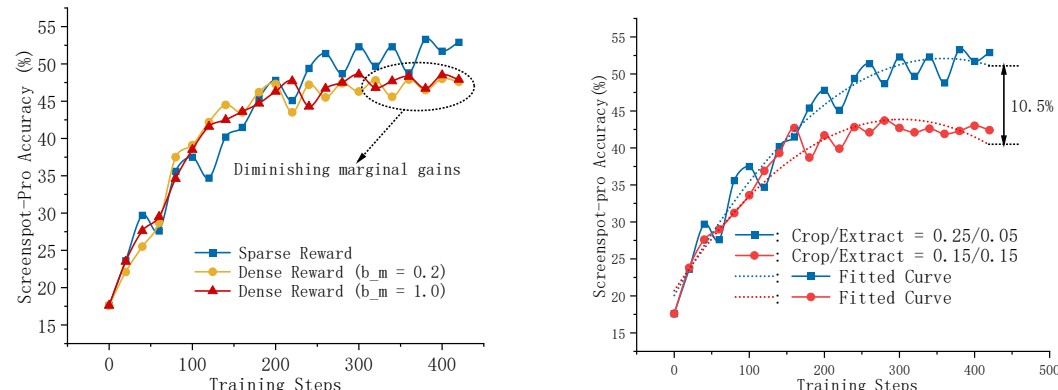

Figure 4: **Left**: Comparison of dense and sparse `Answer` rewards. **Right**: Comparison of different `Crop`/`Extract` reward ratios. b_m: bonus$_{max}$

Based on the experimental results shown in the left panel of Fig. 4, employing a dense `Answer` reward results in marginally lower post-convergence accuracy compared to a sparse `Answer` reward.

We next examine how the relative weighting between the `Crop` and `Extract` rewards affects final performance. As shown in the right panel of Figure 4, moderately increasing the weight of the `Extract` reward relative to the `Crop` reward yields a substantial gain in accuracy. We attribute this to the fact that `Extract` is easier to use than `Crop`: it only requires indicating the approximate location of the target element, without specifying precise coordinates for a bounding box.

## 5 EXPERIMENT

To evaluate the visual grounding capability of GUI-Spotlight, we benchmark it on ScreenSpot-Pro, OSWorld-G, and UI-Vision. The hardware and hyperparameters for training and evaluation, as well as the prompts, are provided in the Appendix A.3.

Table 3: ScreenSpot-Pro evaluation results. Besides GUI-Spotlight, we evaluated UI-TARS-1.5-7B using its official GitHub instructions (UI-TARS-team); results for the other models are taken from the ScreenSpot-Pro leaderboard (Screenspotpro-team). ▨ / ▨: GUI-Spotlight vs. its base.

| Model | Training Data Size | Development | Creative | CAD | Scientific | Office | Operating System | Overall Average |
|---|---|---|---|---|---|---|---|---|
| *Closed-source Models* | | | | | | | | |
| GPT-4o | - | 0.7 | 0.6 | 1.5 | 1.2 | 0.9 | 0.0 | 0.8 |
| Claude Computer Use | - | 12.6 | 16.8 | 11.9 | 25.8 | 26.9 | 8.1 | 17.1 |
| UI-TARS-1.5 | - | - | - | - | - | - | - | 61.6 |
| Seed1.5-VL | - | - | - | - | - | - | - | 60.9 |
| *Open-Source Models 72B Level* | | | | | | | | |
| Qwen2-VL-72B-instruct | - | 1.0 | 0.6 | 0.8 | 2.4 | 0.9 | 0.5 | 1.0 |
| UGround-V1-72B | 10M | 31.1 | 35.8 | 13.8 | 50.0 | 51.3 | 25.5 | 34.5 |
| UI-TARS-72B | - | 40.8 | 39.6 | 17.2 | 45.7 | 54.8 | 30.1 | 38.1 |
| Qwen2.5-VL-72B-Instruct | - | 53.5 | 44.9 | 44.4 | 59.1 | 72.6 | 49.5 | 53.3 |
| GTA-1-72B | 1.56M | 57.2 | 51.0 | 49.8 | 63.0 | 77.0 | 57.1 | 58.4 |
| UI-Venus-72B | 107K | 59.5 | 55.4 | 57.5 | 66.5 | 77.8 | 57.7 | 61.9 |
| *Open-Source Models 32B Level* | | | | | | | | |
| Qwen2.5-VL-32B-Instruct | - | 48.8 | 42.2 | 31.0 | 55.5 | 64.3 | 50.5 | 48.0 |
| GTA-1-32B | 1.56M | 56.2 | 46.3 | 38.7 | 59.1 | 72.2 | 53.1 | 53.6 |
| *Open-Source Models 7B Level* | | | | | | | | |
| See-Click-7B | 1M | 0.3 | 0.6 | 1.9 | 2.0 | 0.9 | 1.5 | 1.1 |
| Qwen2-VL-7B | - | 1.3 | 0.9 | 0.4 | 3.5 | 3.0 | 0.5 | 1.6 |
| UGround-7B | 10M | 14.7 | 17.0 | 11.1 | 19.3 | 27.0 | 9.7 | 16.5 |
| Aguvis-7B | 4.2M | 16.1 | 21.4 | 13.8 | 34.6 | 34.3 | 19.4 | 22.9 |
| Qwen2.5-VL-7B-Instruct | - | 26.1 | 24.0 | 13.0 | 31.1 | 45.2 | 23.5 | 26.8 |
| UGround-V1-7B | 10M | 28.1 | 31.7 | 14.6 | 39.0 | 49.6 | 24.5 | 31.1 |
| UI-TARS-7B | - | 36.1 | 32.8 | 18.0 | 50.0 | 53.5 | 24.5 | 35.7 |
| GUI-Actor-2VL-7B | 9.6M | 38.8 | 40.2 | 29.5 | 44.5 | 56.5 | 36.2 | 40.7 |
| UI-TARS-1.5-7B | - | 33.9 | 33.7 | 25.8 | 47.6 | 63.0 | 33.7 | 38.7 |
| GUI-Actor-2.5VL-7B | 9.6M | 38.1 | 41.3 | 38.3 | 50.8 | 63.0 | 38.8 | 44.6 |
| SE-GUI-7B | 3K | 44.5 | 37.2 | 42.1 | 54.7 | 70.4 | 38.8 | 47.2 |
| GTA-1-7B | 1.56M | 44.5 | 44.0 | 44.4 | 57.1 | 75.2 | 38.3 | 50.1 |
| V2P-7B | 9.6M | 46.8 | 43.1 | 47.1 | 56.3 | 68.3 | 45.4 | 50.6 |
| UI-Venus-7B | 107K | 50.2 | 42.8 | 51.0 | 57.1 | 67.8 | 37.2 | 50.8 |
| *Ours* | | | | | | | | |
| GUI-Spotlight (Init. Qwen2.5-VL-7B-Instruct) | 18.5K↓ | 29.8 | 29.1 | 39.2 | 39.8 | 63.9 | 24.5 | 38.7↑ |
| GUI-Spotlight (Init. UI-TARS-1.5-7B) | 18.5K↓ | 53.3 | 44.4 | 51.0 | 52.4 | 71.3 | 46.9 | 52.8↑ |

## 5.1 HIGH-RESOLUTION PROFESSIONAL GUI GROUNDING

ScreenSpot-Pro (Li et al., 2025) is a benchmark for evaluating visual grounding on high-resolution screenshots of professional software, covering application domains such as creative tools, office platforms and so on. We use it to assess GUI-Spotlight's accuracy on 4K-resolution GUI screens.

As shown in Table 3, our model attains high accuracy, trains data-efficiently, and generalizes broadly. **High accuracy.** GUI-Spotlight (init. UI-TARS-1.5-7B) reaches **52.8**% on SCREENSPOT-PRO, surpassing 7B peers and remaining competitive with much larger models. It improves over its initialization across all six domains, indicating robustness to dense, icon-heavy, cluttered UIs. **Data efficiency.** These results are achieved with only 18.5K curated samples—far less than competing approaches that train on millions (e.g., UGround-V1-7B $\sim$10M, V2P-7B 9.6M). **Generality.** Starting from the non-UI-specific Qwen2.5-VL-7B-Instruct, GUI-Spotlight reaches an absolute +11.9 points over its raw baseline (26.8%), showing that our RL objective and multi-tool coordination transfer beyond UI-specialized backbones and are robust to the choice of backbone.

## 5.2 DESKTOP APPLICATION VISUAL GROUNDING

To evaluate GUI-Spotlight in realistic desktop, we use UI-Vision (Nayak et al., 2025), which pairs diverse screenshots from 83 applications across 6 domains with dense referring expressions.

UI-Vision evaluation results are shown in Table 4, GUI-Spotlight trained from UI-TARS-1.5-7B surpassing its backbone UI-TARS-1.5-7B by +5.3 points and outperforming other 7B models and approaches the 72B UI-TARS-72B. The variant initialized from Qwen2.5-VL-7B attains an absolute gain of +7.4 points over the raw Qwen2.5-VL-7B baseline, evidencing transfer under a non-UI-specific backbone. Overall, these results indicate that our multi-tool RL training consistently improves 7B models and narrows the gap to larger models on UI-Vision.

Table 4: UI-Vision evaluation results. Besides GUI-Spotlight, we evaluated UI-TARS-1.5-7B using its official GitHub instructions (UI-TARS-team); results for the other models are taken from the UI-Venus paper (Gu et al., 2025). ▨ / ▨ : GUI-Spotlight vs. its base.

| Models | Basic | Functional | Spatial | Average |
|---|---|---|---|---|
| *Closed-Source Models* | | | | |
| GPT-4o | 1.6 | 1.5 | 1.0 | 1.4 |
| Claude-3.7-Sonnet | 9.5 | 7.7 | 7.6 | 8.3 |
| *Open-Source Models 72B level* | | | | |
| UI-TARS-72B | 31.4 | 30.5 | 14.7 | 25.5 |
| UI-Venus-Ground-72B | 45.6 | 42.3 | 23.7 | 36.8 |
| *Open-Source Models 7B level* | | | | |
| Qwen2.5-VL-7B | 1.2 | 0.8 | 0.5 | 0.9 |
| OS-Atlas-7B | 12.2 | 11.2 | 3.7 | 9.0 |
| UGround-V1-7B | 15.4 | 17.1 | 6.3 | 12.9 |
| UI-TARS-7B | 20.1 | 24.3 | 8.4 | 17.6 |
| UI-TARS-1.5-7B | 22.9 | 26.1 | 6.6 | 18.1 |
| UI-Venus-Ground-7B | 36.1 | 32.8 | 11.9 | 26.5 |
| *Ours* | | | | |
| GUI-Spotlight (Qwen) | 11.1 | 13.4 | 1.2 | 8.3 |
| GUI-Spotlight (UI-TARS) | 32.1 | 30.2 | 9.1 | 23.4 |

## 5.3 GENERAL-PURPOSE GUI VISUAL GROUNDING

OSWorld-G (Xie et al., 2024) comprises 564 screenshots sourced from OSWorld (Xie et al., 2024), covering a range of operating-system-level tasks such as file operations, application launching, text editing, and system configuration. It emphasizes general-purpose environments where agents must integrate recognition, layout reasoning, and manipulation in everyday workflows.

As shown in Table 5, GUI-Spotlight trained from UI-TARS-1.5-7B achieves an average accuracy of 62.7%, with particularly strong performance on text matching (68.2%) and layout understanding (63.2%). When initialized from Qwen2.5-VL-7B, the model raises the average score from 31.4% to 35.6%, gaining substantially in element recognition (+17.3) and showing smaller improvements in text matching (+1.7) and manipulation (+2.1), with only a modest drop in layout understanding (−1.8). These results indicate that reinforcement learning with tool-augmented feedback provides clear benefits even when starting from a non-UI-specific backbone. Moreover, despite being trained on far fewer examples, the 7B-scale GUI-Spotlight remains competitive with 72B-scale models, supporting its robustness for diverse OS-level grounding tasks.

Table 5: OSWorld-G evaluation results. Besides GUI-Spotlight, we evaluated UI-TARS-1.5-7B using its official GitHub instructions (UI-TARS-team); results for the other models are taken from the UI-Venus Technical Report (Gu et al., 2025). ░ / ░ : GUI-Spotlight vs. its base.

| Models | Text Matching | Element Recognition | Layout Understanding | Fine-grained Manipulation | Refusal | Average |
|---|---|---|---|---|---|---|
| *Closed-Source Models* | | | | | | |
| Operator | 51.3 | 42.4 | 46.6 | 31.5 | - | 40.6 |
| Gemini-2.5-pro | 59.8 | 45.5 | 49.0 | 33.6 | 38.9 | 45.2 |
| Seed1.5-VL | 73.9 | 66.7 | 69.6 | 47.0 | 18.5 | 62.9 |
| *Open-Source Models-72B Level* | | | | | | |
| UI-TARS-72B | 69.4 | 60.6 | 62.9 | 45.6 | - | 57.1 |
| Qwen2.5-VL-72B | 52.6 | 74.6 | 74.7 | 55.3 | - | 62.2 |
| UI-Venus-Ground-72B | 82.1 | 71.2 | 70.7 | 64.4 | - | 70.4 |
| *Open-Source Models-7B Level* | | | | | | |
| OS-Atlas-7B | 44.1 | 29.4 | 35.2 | 16.8 | 7.4 | 27.7 |
| Qwen2.5-VL-7B | 45.6 | 32.7 | 41.9 | 18.1 | - | 31.4 |
| UGround-7B | 51.3 | 40.3 | 43.5 | 24.8 | - | 36.4 |
| Aguvis-7B | 55.9 | 41.2 | 43.9 | 28.2 | - | 38.7 |
| UI-TARS-7B | 60.2 | 51.8 | 54.9 | 35.6 | - | 47.5 |
| Jedi-7B | 65.9 | 55.5 | 57.7 | 46.9 | 7.4 | 54.1 |
| UI-Venus-Ground-7B | 74.6 | 60.5 | 61.5 | 45.5 | - | 58.8 |
| UI-TARS-1.5-7B | 67.3 | 64.5 | 65.2 | 42.9 | - | 61.9 |
| GTA1-7B | 63.2 | 82.1 | 74.2 | 70.5 | - | 67.7 |
| *Ours* | | | | | | |
| GUI-Spotlight (Init. Qwen2.5-VL-7B) | 47.3 | 50.0 | 40.1 | 20.2 | - | 35.6 |
| GUI-Spotlight (Init. UI-TARS-1.5-7B) | 68.2 | 60.6 | 63.2 | 45.6 | - | 62.7 |

## 5.4 GUI-Spotlight vs. training-free Iterative Inference

GUI-Spotlight performs multi-step reasoning at inference time. To quantify its gains over training-free iterative inference, we conduct an ablation study comparing GUI-Spotlight with two baselines: ① Multi-turn conversational inference: we use the same multi-round tool prompts as GUI-Spotlight and, after each turn, append the executed tool outputs to the dialogue history. ② Repeated single-turn inference: following the setting in InfantAgent-Next (Lei et al., 2025) for vision models, after the first attempted click we crop a $700 \times 450$-pixel region centered at the predicted coordinates and continue issuing clicks within this region in subsequent attempts.

Our results show that the model initially has virtually no multi-step reasoning capacity. After training, however, the multi-step reasoning model attains higher accuracy than a baseline that iterates single-turn steps—at each step it performs one click, crops a local region centered on that click, feeds the cropped image back into the model, and then re-locates on the crop, repeating this procedure multiple times. This demonstrates a substantive post-training gain in GUI-Spotlight.

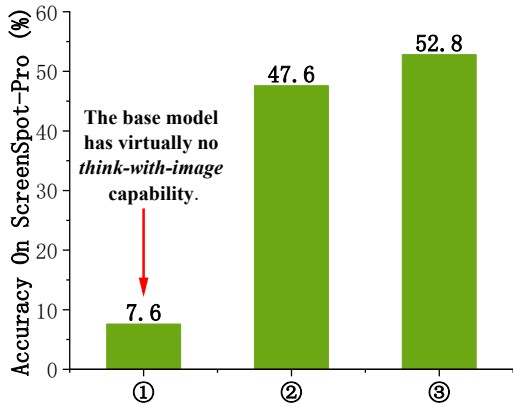

Figure 5: Comparison of multi-step reasoning strategies. UI-TARS-1.5-7B is used as the initial model: ① multi-turn conversational inference; ② repeated single-turn inference; ③ GUI-Spotlight.

## 6 Conclusion

We introduced GUI-Spotlight, a *think-with-image* visual grounding model that coordinates multiple tools through a stabilized GSPO-based reinforcement learning procedure. With only $18.5K$ training samples, it attains $52.8\%$ on ScreenSpot-Pro and $23.4\%$ on UI-Vision, remaining competitive with substantially larger models. Beyond raw accuracy, our multi-tool RL design improves training stability and sample efficiency, and our comprehensive documentation (including negative results) offers practical guidance for building agentic grounding models with coordinated tool use.

ETHICS STATEMENT

This work does not involve human subjects, personal data, or sensitive attributes. All experiments are conducted on publicly available or newly collected GUI screenshots, where the collection process relies on automated crawling and synthetic generation of interface states. We took care to filter out inappropriate or potentially harmful content during dataset creation. The methodology focuses exclusively on improving visual grounding for GUI agents and does not introduce foreseeable ethical risks beyond standard concerns of bias inherent in large models.

REPRODUCIBILITY STATEMENT

We have taken multiple steps to ensure reproducibility. Details of the datasets, data cleaning pipeline, training stages, and learning objectives are described in Section 3.2, reinforcement learning objectives in Section 4.1, and reward design in Section 4.2. Hyperparameters and training settings are provided in Appendix A. Due to the large size of the full model and datasets, they cannot be included directly at submission time, but will be released publicly once the paper is published. Additionally, we thank the UGround team for open-sourcing their dataset. As described in our paper, we filter their dataset to construct our training data.

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

## A   APPENDIX

### A.1   THREE STAGES TRAINING HYPERPARAMETER

The specific parameters used in the first stage of training are listed in Table 6. The parameters used in the second and third stage of training are listed in Table 7.

Table 6: Training hyperparameters for Stage 1

| Hyperparameter | Value |
|---|---|
| Finetuning type | full |
| Freeze vision tower | true |
| Freeze multimodal projector | true |
| Freeze language model | false |
| Precision | bf16 |
| Learning rate | 1.0e-5 |
| LR scheduler | cosine |
| Warmup ratio | 0.1 |
| Per-device batch size | 12 |
| Gradient accumulation steps | 2 |
| Effective global batch | $12 \times 2 \times N_{\text{GPU}}$ |
| Training epochs | 1.0 |
| Max sequence length | 10000 |
| Distributed/parallel | DeepSpeed ZeRO-3 |

### A.2   TRAINING PARAMETERS FOR ALGORITHMIC EXPLORATION

The specific training parameters for algorithmic exploration are listed in Table 8. By default, we use $\epsilon = 0.2$. In the *clip-higher* setting, we set $\epsilon_{\text{low}} = 0.2$ and $\epsilon_{\text{high}} = 0.28$. The KL coefficient is $\beta_{\text{KL}} = 0.01$ by default, and $\beta_{\text{KL}} = 0$ in the *no-KL* ablation.

### A.3   TRAINING AND EVALUATION DETAILS FOR EXPERIMENTS

**Hardware** Experiments were conducted on a multi-GPU server with 8 NVIDIA H200 (144 GB HBM3e each), interconnected via NVLink (NV18 links across all pairs. The GPUs ran with

Table 7: Stage 2 and 3 training hyperparameters.

| Hyperparameter | Value |
|---|---|
| Learning rate | 1e-6 |
| LR scheduler | constant_with_warmup |
| Warmup steps | 10 |
| Training epochs | 1 |
| Temperature | 1.0 |
| Clip range $\epsilon$ | 0.2 |
| Clip range high $\epsilon_{\text{high}}$ | 0.28 |
| Precision | bf16 |
| Max grad norm | 0.01 |
| Iterations per run | 2 |
| KL coefficient $\beta$ | 0.00 |
| Max prompt length | 1024 |
| Max completion length | 4096 (stage 2)/15000 (stage 3) |
| Per-device train batch | 6 |
| Gradient accumulation steps | 16 |
| Num generations | 6 |

Table 8: Algorithmic exploration training hyperparameters.

| Hyperparameter | Value |
|---|---|
| Learning rate | 1e-6 |
| LR scheduler | constant_with_warmup |
| Warmup steps | 10 |
| Training epochs | 1 |
| Temperature | 1.0 |
| Precision | bf16 |
| Max grad norm | 0.01 |
| Iterations per run | 2 |
| Max prompt length | 1024 |
| Max completion length | 4096 (stage 2)/15000 (stage 3) |
| Per-device train batch | 6 |
| Gradient accumulation steps | 16 |
| Num generations | 6 |

NVIDIA driver 575.57.08 and CUDA 12.9 (MIG disabled, persistence mode enabled). The host is a dual-socket AMD EPYC 9454 machine (248 cores, 192 threads total) with 1.5 TiB system memory. We conducted all training and evaluation on GPUs 4–7.

**Hyperparameters** The training phase used the same hyperparameters as the three-stage training procedure (Appendix A.1). For evaluation, we employed the vLLM library; specifically, we set tensor_parallel_size=1, gpu_memory_utilization=0.95, max_model_len=30000, max_tokens=19263, temperature=0, top_p=1.0, and a batch size of 64.

**Prompts** We consistently use the following prompt for both training and evaluation.

Listing 1: Tool_Prompt

```
You should use three tools to help you analyze the image and find the
    target coordinate:
1. **crop**: This tool allows you to crop a specific area of the image by
    specifying the top-left and bottom-right coordinates of the
    rectangle you want to crop.
2. **extract**: This tool allows you to extract one quarter of the image
    based on the specified horizontal and vertical positions (left,
    center, right for x-axis; top, center, bottom for y-axis).
3. **find_color**: This tool allows you to find a specific color in the
    image by providing the RGB values of the target color.
Example Usage:
```

```
<crop>(Image_0, (10, 20), (110, 100))</crop> # Crop a rectangle from
    Image_0 from (10, 20) to (110, 100)
<extract>(Image_0, left, top)</extract> # Extract the top-left quarter of
     Image_0
<find_color>(Image_2, (255, 0, 0))</find_color> # Find the red color in
    Image_2
Before each tool call, please enclose your reasoning within <think>...</
    think> tags.
In the end, you should return your final answer using the <answer>...</
    answer> tag.
In the <answer> tag, you should return the image and the coordinate of
    the target object in the format (Image_X, (x, y)), where Image_X is
    the image containing the target object and (x, y) is the coordinate
    of the target object.
Here is an example of how to find the final target coordinate:
User: [Image_0 is displayed above] Could you identify the location of the
     Close button in this interface?

Assistant: <think>I would like to zoom in on the menu icon at the top-
    left corner to get a clearer view.</think>
<crop>(Image_0, (10, 20), (110, 100))</crop>

User: [Image_1 is displayed above] # (cropped image returned)xw

Assistant: <think>In this cropped image, I cannot see the Close button; I
     will use the `extract` tool to find the possible area where the Close
     button is located.</think>
<extract>(Image_0, left, top)</extract>

User: [Image_2 is displayed above] # (possible area returned)

Assistant: <think>In this cropped image, I can see the approximate
    position of the Close button, it sits near the center of the region,
    slightly toward the lower-right. It is at (45, 60).</think>
<answer>(Image_2, (45, 60))</answer>

Now, let's work on the real task:
[Image_0 is displayed below]
```

### A.4 HIGH-RESOLUTION WEB GUI DATASET CONSTRUCTION

***Data sources and domain selection:*** We build a small but diverse high-resolution subset using an automated, browser-based collection pipeline on top of headless Chrome. We manually curate a list of high-traffic websites that cover several major usage domains, including search engines, social media, messaging, e-commerce, content platforms, and utilities (e.g., google.com, youtube.com, facebook.com, instagram.com, chatgpt.com, wikipedia.org, reddit.com, x.com, amazon.com, net-flix.com, temu.com, etc.). For each domain in this list, we launch a headless Chrome instance with a fixed $3840 \times 2160$ viewport, navigate to `https://<domain>`, wait for the page to finish loading, and capture a full-screen screenshot.

***Element discovery and filtering:*** On each loaded page, we detect candidate interactive elements via a unified XPath that selects (i) anchor tags `<a>`, (ii) `<button>` elements, (iii) elements with an `onclick` handler, and (iv) elements with accessibility roles such as `role="button"` or `role="search"`. We then apply several additional functions to filter out low-quality candidates:

1. **Viewport visibility.** Using `getBoundingClientRect` in JavaScript, we check that an element is fully inside the current viewport (its bounding box must lie within $[0, 3840] \times [0, 2160]$). Elements that are off-screen or only partially visible (e.g., require scrolling) are discarded.

2. **Size threshold.** We remove elements whose rendered bounding box is tiny or likely invisible, by requiring both width and height to be at least 5 pixels.

3. **Meaningful label.** For each remaining element, we extract a textual label as supervision. We first read the visible text (`elem.text`), and if it is empty, we fall back to the `aria-label` attribute. Elements with neither visible text nor an accessibility label are dropped, ensuring that each retained element has a semantically meaningful description.

***Bounding box extraction and alignment to the screenshot:*** For every element that passes the above filters, we obtain its location and size directly from the browser's layout engine via Selenium (`elem.location` and `elem.size`), which internally correspond to the top-left coordinates and width/height returned by `getBoundingClientRect()` in CSS pixels. Since the browser runs in a fixed $3840 \times 2160$ viewport and we capture the screenshot of exactly this viewport, the coordinate system of the full-screen image is aligned one-to-one with these CSS pixel coordinates (under the default device pixel ratio). Concretely, if an element has location $(x, y)$ and size $(w, h)$, we crop the rectangle $(x, y, x + w, y + h)$ from the full screenshot. This yields a tightly aligned element crop for each DOM element without any manual calibration or heuristic rescaling.

Each crop is saved as an individual PNG file, and we write out a JSON entry containing the element's text label, its bounding box on the original $3840 \times 2160$ screenshot, and the path to the cropped image. In this way, every sample in the high-resolution subset consists of a real-world clickable GUI element paired with its on-screen coordinates.

***Instruction construction:*** For this crawled subset, we derive natural-language instructions directly from the collected element labels using a small set of templates. Concretely, given a text label $L$ (e.g., "Search", "Sign in", "Play"), we instantiate instruction prompts such as "Click the **'L'** button." or "Find and click **'L'**." This yields (instruction, full screenshot, target bounding box) triples without relying on any proprietary data source. We will clarify this instruction generation step in the revised version.

## USE OF LARGE LANGUAGE MODELS (LLMS)

Large language models were used in two aspects of this work: (1) as backbones for initialization (e.g., UI-TARS-1.5-7B, Qwen2.5-VL-7B), and (2) as evaluators in the data filtering pipeline, where Qwen2.5-VL-72B was used to audit instruction quality and annotation correctness. For manuscript writing, LLMs were used as assistive tools for language refinement, but all content was reviewed, verified, and finalized by the authors. LLMs were not involved in ideation or experimental design and are not eligible for authorship.

