# OpenReview forum: "GUI-Spotlight: Adaptive Iterative Focus Refinement for Enhanced GUI Visual Grounding"
_ICLR.cc/2026/Conference — Submitted to ICLR 2026_

### Official Review · Reviewer_5C7T · 2025-10-26

**Soundness:** 3
**Presentation:** 3
**Contribution:** 3
**Rating:** 2
**Confidence:** 5

**Summary:**

This paper mainly focused on the GUI Grounding task. The authors proposed an adaptative iterative focus refinement method to progressively encourage models using tools, e.g., cropping sub-regions, for more fine-grained information and enhance the overall grounding accuracy. A modified Group Sequence Policy Optimization (GSPO) algorithm was proposed to help MLLM adapt to the tool using mechanism. Multiple experiments are presented to show the effectiveness of the proposed method.

**Strengths:**

The idea of encouraging models to use tools to progressively ground potential GT elements is interesting.

**Weaknesses:**

1. Table 1 shows three available tool functions and I have some questions
  - extract can be considered a special case of crop. So I wonder why is the extract function included?
  - what is the motivation for including the find_color function? Was it because MLLMs were found to have errors in color recognition?
  - If there are multiple regions matching the target color, how are the offsets for these multiple regions represented in the return value, and how are the sizes of the corresponding regions indicated?
2. In Section 3.2.2, Stage 1, it is stated that "collected 2561 multi-turn dialogue trajectories with tool invocations." Please present more detail of how this multi-turn dialogue data incorporating tool usage acquired.
3. The proposed method for element grounding requires inference involving multiple tool calls, which will significantly increase the time and computational overhead for the grounding task. Have the authors statistically analyzed the average number of tool calls required to complete a single grounding task? Could the authors provide a comparison of the time and computational overhead against existing methods? Furthermore, in the context of a GUI Agent, this overhead would severely impact the real-time performance of the agent.
4. Based on the tool use prompt template shown in Appendix A.3, the model's need to call a tool is likely because the input image resolution is too large (considering computational costs, the MLLM performs some degree of compression), and the MLLM expects to zoom in on a local area to see more clearly. For the extract/crop functions, after obtaining the local region, is the sub-region image enlarged before being fed back into the model? If so, to what size is it enlarged? On the other hand, providing only a local region might lead to incomplete context, which is problematic since the grounding for some tasks relies on surrounding context. How does the proposed method mitigate this issue of context loss?
5. For the GUI Grounding task, some current researches are exploring more efficient coordinate-free grounding methods, such as Attention-driven GUI Grounding: Leveraging Pretrained Multimodal Large Language Models without Fine-Tuning (https://arxiv.org/abs/2412.10840) and GUI-Actor: Coordinate-Free Visual Grounding for GUI Agents (https://arxiv.org/abs/2506.03143). GUI-Actor, based on Qwen-2-VL-7B-Instruct, achieved a performance of 40.9\% on ScreenSpot-Pro, while the proposed method, based on Qwen2.5-VL-7B-Instruct, only achieved 38.7\% in Table 3. Furthermore, the inference time and computational overhead of the proposed method might be higher. It is suggested that the authors provide a discussion on this point, explaining the necessity of the proposed multi-turn iterative grounding pipeline.
6. This paper lacks exploration of GUI Agent Systems. GUI grounding is not an isolated task; it is a critical component within a comprehensive GUI Agent System. MLLM-based GUI Agent Systems aim to increase the success rate of complex task completion by improving grounding quality, ideally through an economical, efficient, and unified integration into the MLLM inference process. This paper exclusively explores MLLM-based GUI grounding but fails to investigate the crucial next step: how to seamlessly integrate this capability into a functional GUI Agent System is a critical step.

**Questions:**

Please find the question in the Weaknesses section.

---

> ### Author Response · Authors · 2025-11-25
>
> We thank the reviewer for the thoughtful and constructive feedback. Below, we provide point‑by‑point responses to the comments.
>
> ----
>
> For ***Weaknesses 1:***
>
> > extract can be considered a special case of crop. So I wonder why is the extract function included?
>
> our goal in designing the tool set is to support a **coarse-to-fine localization strategy** while keeping the tools as simple and learnable as possible for the model. Concretely, `extract` is a very simple, structured operation: it always crops a **fixed 1/4 sub-region** of the original image (height and width both halved), specified by a discrete position (top-left, top-right, bottom-left, bottom-right). This design makes it easy for the model to first remove large irrelevant areas and focus on a salient region with **minimal coordination overhead**—the model does not need to reason about precise bounding-box coordinates to use `extract` effectively.
>
> In contrast, `crop` requires the model to specify an explicit bounding box, which is **much more general but also harder to learn** (the model must predict continuous or fine-grained coordinates). At the same time, `crop` can handle cases that `extract` cannot, such as very elongated elements (e.g., long search bars) that are not well captured by a fixed 1/4 region. In this sense, the two tools are **complementary**:
> - `extract` is **simple and easy to learn**, but not fully general;
> - `crop` is **fully general**, but requires more precise spatial reasoning.
>
> We found this combination to work well in practice: the model typically uses `extract` for coarse localization and falls back to `crop` when finer control over the region is needed.
>
> > what is the motivation for including the find_color function? Was it because MLLMs were found to have errors in color recognition?
>
> The `find_color` tool is designed for cases where the target element has a distinctive color but is hard to localize precisely. For example, consider a very small, bright red “close” button on the screen: due to its tiny size, the model’s direct localization may be imprecise. In contrast, calling `find_color` allows the agent to accurately locate such color-salient elements in a single step.
>
> > If there are multiple regions matching the target color, how are the offsets for these multiple regions represented in the return value, and how are the sizes of the corresponding regions indicated?
>
> As summarized in **Table 1 and Section 3.1 Tool Functions part** , `find_color` is designed to return **a single focused region**, not a list of all matches. Concretely, the tool:
>
> 1. Scans the original image with **10×10 patches** (stride 10).
> 2. For each patch, computes the perceptual color difference ΔE in CIE Lab space to the target RGB.
> 3. Selects the patch with **minimal ΔE** (i.e., the best color match over the whole screen).
> 4. Centers a fixed **\(w_s \times w_s\)** crop window around that patch.
>
> The output of `find_color` is therefore a single cropped image together with an offset \((\Delta x, \Delta y)\), which denotes the **top-left corner of this \(w_s \times w_s\) crop relative to the original image** (or `None` if no valid crop can be formed). In cases where multiple regions share a similar color, the tool implicitly resolves this by choosing the region whose average color is **closest** to the target in ΔE; we do **NOT** return multiple offsets simultaneously.
>
> The returned crop has a fixed size of \(w_s \times w_s\)
> , and the corresponding offset in the original image is directly computed by the tool and written back into the model’s dialogue history.

---

> > ### Author Response · Authors · 2025-11-25
> >
> > For ***Question 2:***
> >
> > > Please present more detail of how this multi-turn dialogue data incorporating tool usage acquired.
> >
> > As described in **Section 3.2.2**, we *first executed the same inference pipeline with Qwen2.5-VL-72B on the filtered UGround dataset and collected 2,561 multi-turn dialogue trajectories with tool invocations.* The concrete procedure for constructing this dataset is as follows:
> >
> > 1. **Tool-augmented inference setup.**
> >    We exposed the same three tools (`extract`, `crop`, `find_color`) to **Qwen2.5-VL-72B** by providing natural-language descriptions and function signatures in the system prompt. The model was instructed that it could interleave normal responses with tool calls to solve grounding tasks.
> >
> > 2. **Generating trajectories on UGround.**
> >    For each image–instruction pair in the filtered UGround dataset, we ran an auto-regressive rollout with Qwen2.5-VL-72B using this tool-augmented interface. During the rollout, the model could:
> >    - call a tool (with arguments),
> >    - receive the tool’s return (cropped image + offset message),
> >    - continue reasoning based on the updated dialogue history,
> >    until it produced a final grounding prediction or reached a small step limit.
> >
> > 3. **Forming multi-turn dialogues.**
> >    Each rollout naturally yields a **multi-turn dialogue trajectory** consisting of:
> >    - model utterances,
> >    - explicit tool calls,
> >    - tool responses (including offsets and info messages),
> >    and the final answer. These trajectories are exactly the format used later for Stage-1 SFT.
> >
> > 4. **Filtering for high-quality tool usage.**
> >    We then filtered these rollouts by:
> >    - keeping only cases where the final grounding matched the UGround ground truth, and
> >    - restricting to trajectories that used tools **1–2 times** (to avoid degenerate behaviors with many unnecessary calls).
> >
> >    After this filtering, we collected **2,561** high-quality multi-turn trajectories with tool invocations, which are used as the Stage-1 SFT data.
> >
> > For ***Question 3:***
> >
> > > Have the authors statistically analyzed the average number of tool calls required to complete a single grounding task?
> >
> > Our main goal with GUI-Spotlight is to **improve grounding accuracy** via a *think-with-image* procedure: the model performs a small number of tool-augmented visual reasoning steps before committing to a final answer. As summarized in Table below, we quantified this behavior across benchmarks.
> >
> > | Benchmark      | # Cases | Cases with 1 tool call | Cases with 2 tool calls | Avg. tool calls / case |
> > |:--------------:|:-------:|:----------------------:|:-----------------------:|:----------------------:|
> > | ScreenSpot-Pro |  $1581$   |         $1498$           |           $83$            |         $1.05$           |
> > | UI-Vision      |  $1460$   |         $1407$           |           $53$            |         $1.04$           |
> >
> > In other words, GUI-Spotlight almost always performs **only a single tool-augmented step** before answering, and cases with more than two calls are essentially nonexistent in our current setting. Each call corresponds to a forward pass on a **cropped sub-region** plus a short textual update, so the overhead is \(O(1)\) extra vision–language evaluations per example, rather than a long multi-step interaction. It is worth noting that the **cropped images are much smaller than the original screenshots, so they consume far fewer image tokens**. As a result, the overall inference cost does not increase significantly.
> >
> > This modest overhead is balanced by a substantial accuracy gain. When initialized from the same **UI-TARS-1.5-7B** checkpoint, the base model achieves $38.7$% on ScreenSpot-Pro, whereas GUI-Spotlight, with the same backbone and only $18.5$K additional training examples, improves performance to $52.8$%. We view this ~$14$-point absolute improvement as a favorable trade-off for the small additional cost of ~$1$ tool interaction per case.
> >
> > In the context of a full GUI agent, this pattern is particularly important: the agent does **not** repeatedly call tools in an unbounded loop; instead, it typically performs a single “focus” step (via our tools) and then answers. Thus, our method behaves as a low-iteration refinement layer on top of existing grounding models.

---

> > > ### Author Response · Authors · 2025-11-25
> > >
> > > For ***Question 4:***
> > >
> > > >  For the extract/crop functions, after obtaining the local region, is the sub-region image enlarged before being fed back into the model?
> > >
> > > Our main motivation for using tools is **not** to compensate for resolution compression, but to help the model **remove irrelevant regions** and focus on the area around the target element. In other words, tool calls act as a lightweight *focusing* mechanism: they prune away large parts of the screen that are clearly unrelated to the query, making subsequent grounding easier and more reliable.
> > >
> > > For the sub-region processing:
> > > - The images returned by `extract` and `crop` are **not resized or enlarged** in any way.
> > > - The tool simply returns the selected sub-image at its original resolution; the goal is purely to **discard irrelevant content**, not to upsample or change the scale.
> > >
> > > > On the other hand, providing only a local region might lead to incomplete context, which is problematic since the grounding for some tasks relies on surrounding context. How does the proposed method mitigate this issue of context loss?
> > >
> > > Regarding potential context loss:
> > > - First, as discussed in response to question 1, the model has access to both `extract` and `crop`. It can use `extract` for coarse focusing and `crop` when a larger or more precisely shaped region is needed, which helps preserve sufficient local context around the target.
> > > - More importantly, the **original full image remains in the dialogue history throughout inference** (it is always present as the initial visual input). Thus, the model can always refer back to the global view when needed; the tools add *extra* localized views rather than replacing the original image.
> > > - In our response to **reviewer yySw**, we also provide a schematic illustration (**second message in that discussion**) that visualizes this process: the full image is kept as global context, and tool calls progressively add focused sub-images as additional evidence, rather than discarding the original context.
> > >
> > > For ***Question 5:***
> > >
> > > > GUI-Actor, based on Qwen-2-VL-7B-Instruct, achieved a performance of 40.9% on ScreenSpot-Pro
> > >
> > > Thank you for raising this point. It is important to note that GUI-Actor is trained on a **much larger** amount of data: according to their paper, the action head is supervised on about $9.6$M GUI elements from ~$1$M screenshots, in addition to a separate $730$K-example verifier dataset. In contrast, our method uses only $18.5$K training examples for GUI visual grounding—roughly **$1/500$** of the data scale used by GUI-Actor.
> > >
> > > As a result, we view our approach as a **data-efficient, multi-turn tool-using pipeline** that is complementary to coordinate-free methods like GUI-Actor, rather than a direct competitor under the same data regime. Our focus is on teaching a GUI agent to perform **iterative reasoning and tool usage** with minimal additional supervision, while GUI-Actor demonstrates what is achievable with a substantially larger, specialized grounding corpus.
> > >
> > > We will include these discussions and cite their work in the final version.
> > >
> > > For ***Question 6:***
> > >
> > > >This paper lacks exploration of GUI Agent Systems
> > >
> > > Thank you for this suggestion.
> > >
> > > To specifically measure the impact of grounding on agent performance while **factoring out planner effects**, the OSWorld team released the **OSWorld-G** benchmark.
> > >
> > > To evaluate whether GUI-Spotlight can enhance an agent’s accuracy on real-world GUI tasks, we follow the [Qwen3-VL](https://camo.githubusercontent.com/7b30ae6af5402a71bc63076de6e1b67d5bf94c1ada1c3fb59fddaf30b98fbd8a/68747470733a2f2f7169616e77656e2d7265732e6f73732d616363656c65726174652e616c6979756e63732e636f6d2f5177656e332d564c2f7461626c655f6e6f7468696e6b696e675f766c2e6a7067) evaluation setup and measured the GUI-Spotlight's performance on OSWorld-G, as reported in **Section 5.3** of our paper.
> > >
> > > In addition, we conducted an agent evaluation using the [InfantAgent](https://arxiv.org/abs/2505.10887) framework, with Claude 4.5 sonnet as the planner and either **GUI-Spotlight** or **UI-TARS-1.5-7B** as the GUI grounding module. Over 50 steps, replacing the original UI-TARS-1.5-7B grounding with GUI-Spotlight yields a modest but consistent improvement in task success rate:
> > >
> > > | Grounding model            | Agent success rate on OSWorld (%) |
> > > |:--------------------------:|:----------------------------------:|
> > > | UI-TARS-1.5-7B (baseline)  |              $58.17$                 |
> > > | GUI-Spotlight              |              $60.90$                 |
> > >
> > >
> > > These results demonstrate that our **multi-turn iterative grounding pipeline can be seamlessly integrated into a functional GUI agent system** and yields consistent gains on downstream agent tasks, beyond static ScreenSpot-Pro / UI-Vision benchmarks.
> > >
> > > ----
> > >
> > > Once again, thank you for all of your thoughtful questions, and we look forward to your further feedback.

---

### Official Review · Reviewer_Kbre · 2025-10-28

**Soundness:** 3
**Presentation:** 2
**Contribution:** 2
**Rating:** 4
**Confidence:** 3

**Summary:**

This paper proposes GUI-Spotlight, a reinforcement learning framework for GUI grounding that coordinates multiple tools (crop, extract, find_color) to enable multi-step “think-with-image” reasoning. The approach fine-tunes 7B vision-language models with 18.5K examples using a GSPO-based algorithm to stabilize training. Experiments on ScreenSpot-Pro (52.8%) and UI-Vision (23.4%) show that GUI-Spotlight achieves competitive performance compared with larger models.

**Strengths:**

**1. Careful Experimental Analysis**: The paper conducts extensive ablations on RL algorithms (Section 4.1) and reward design (Section 4.2), providing valuable insights beyond reporting final results. The inclusion of negative results is particularly commendable.

**2. Comprehensive Empirical Results**: Consistent improvements across multiple benchmarks (ScreenSpot-Pro: 52.8%, UI-Vision: 23.4%, OSWorld-G: 62.7%) demonstrate robustness. The method works well under different initializations (Qwen2.5-VL-7B and UI-TARS-1.5-7B).

**3. Practical System Design**: The tool-based iterative refinement pipeline is intuitive and interpretable. The three tools (*crop*, *extract*, *find_color*) are simple and easy to use, and the inference pipeline is clear and implementable.

**4. Rigorous Data Curation**: The multi-stage filtering pipeline, which leverages Qwen2.5-VL-72B for instruction quality checks, bounding box validation, and consistency filtering, ensures high-quality training data.

**5. Training Stability Innovation**: The modified GSPO with auxiliary loss J'(θ) and tool-filtered sampling addresses key challenges in multi-turn RL training and effectively prevents collapse observed in vanilla GRPO/GSPO.

**Weaknesses:**

**1. Lack of Comparison with Recent Baselines**

The evaluation on ScreenSpot-Pro does not include several stronger 7B baselines that are already reported on the leaderboard, such as GUI-ARP-7B (60.8%), Holo1.5-7B (57.9%), GTA1-7B (55.5%), and GUI-Cursor (56.5%). Although some of these works may be concurrent, it would strengthen the work to demonstrate whether the proposed iterative tool-coordination approach can further improve these stronger base models, helping to isolate the contribution of the methodology versus the initialization choice.

**2. Data Efficiency Claims**

The claim of achieving results with "only 18.5K samples" could benefit from more clarification. The model initializes from UI-TARS-1.5-7B, which was pre-trained on millions of GUI examples, so the 18.5K represents incremental fine-tuning rather than training from scratch. When starting from the non-GUI-specific Qwen2.5-VL-7B baseline, performance drops to 38.7% on ScreenSpot-Pro, which still lags behind other grounding models trained on large-scale datasets. Clarifying the relative contributions of the base model versus the proposed training procedure would better position this work as an incremental improvement built upon strong pre-training.

**3. Insufficient Analysis of RFT Design Choices**

The paper employs a modified GSPO algorithm but lacks detailed comparison with other recent RFT approaches. For instance, UI-AGILE uses only 9K examples (versus 18.5K in this work) and achieves 48.7% on ScreenSpot-Pro when initialized from Qwen2.5-VL, compared to 38.7% reported here with the same baseline.

**4. Simplistic Tool Design**

The tool set (crop, extract, find_color) is relatively simple. Given that GUIs are not natural images but structured environments with explicit hierarchies, element types, and interaction semantics, incorporating structure-aware approaches (e.g., pretrained GUI element detectors, OCR-based text localization) could potentially improve both efficiency and accuracy. The current design treats GUIs as generic images, which may limit extensibility to more complex tasks.

**5. Presentation Issues**

- **Citation formatting**: Inconsistent use of `\citep` vs. `\citet` throughout. For example, lines 81-90 in Section 2 incorrectly format parenthetical citations as narrative ones (e.g., "...screen coordinates You et al. (2024)" should be "...screen coordinates \citep{You2024}"), creating inconsistent formatting across references.

- **Layout**: Some figures and tables have insufficient spacing, which affects readability. For instance, the gap between Figure 4 and its caption is too small, and in Section 5.2, the spacing below Table 4 is too tight against the heading of Section 5.3. Adjusting the vertical spacing would improve the overall visual clarity.

- **Minor inconsistency**: Mixed use of "GUI-SPOTLIGHT" versus "GUI-Spotlight". Using one form consistently would be clearer.

These issues do not affect the technical content but could be easily addressed in revision to improve clarity and readability.

**Questions:**

**1. Ablation on Stronger Baselines**

Can the proposed iterative tool-coordination approach be applied to stronger 7B GUI grounding models such as GUI-ARP-7B, Holo1.5-7B, GTA1-7B, and GUI-Cursor-7B? Demonstrating improvements on these baselines would help strengthen the contribution and better distinguish the method from base-model choices.

**2. Comparison with Other RFT Methods**

For example, UI-AGILE achieves 48.7% on ScreenSpot-Pro with only 9K samples using Qwen2.5-VL as the base, compared to 38.7% reported here with 18.5K samples. What accounts for this discrepancy? Are there complementary design choices from UI-AGILE or other RFT methods that could be integrated into GUI-Spotlight?

**3. Tool Usage Ablation**

What is the distribution of tool invocations during inference? How many tool calls are typically needed before producing a final answer, and does this vary across benchmarks or UI complexity levels? An ablation on the usage of individual tools could provide insights into the model’s reasoning patterns and inform future tool design.

**4. Tool Generalization**

Have the authors considered extending the tool set to leverage GUI-specific structure? For example, incorporating GUI element detectors or OCR-based text localization could potentially reduce iteration steps and improve efficiency on more complex interfaces.

**5. Presentation Issues**

Fixing the presentation issues noted in the weaknesses would improve clarity and readability.

---

> ### Author Response · Authors · 2025-11-25
>
> We thank the reviewer for the thoughtful and constructive feedback. Below, we provide point‑by‑point responses to the comments.
>
> ----
>
> For ***Weaknesses 1 & Questions 1:***
>
> > Lack of Comparison with Recent Baselines
>
> Thank you for pointing this out. However, the models you mentioned were all released after the ICLR 2026 submission deadline (Sep 19, 2025), so it was not possible for us to include them in the original submission:
>
> - **GUI-ARP-7B** was released on **Sep 19, 2025**
>   ([arXiv:2509.15532](https://arxiv.org/abs/2509.15532)).
>
> - **GUI-Cursor-7B** was released on **Sep 25, 2025**
>   ([arXiv:2509.21552](https://arxiv.org/abs/2509.21552)).
>
> - **Holo1.5**: the technical report describing Holo1.5 was posted on **Oct 9, 2025**
>   ([blog post](https://www.hcompany.ai/blog/holo-1-5)).
>
> - **GTA1-7B**: two model versions were uploaded.
>   We already compare against the **first version** in **Table 3** of our submission.
>   The **second version** was committed on **Oct 1, 2025**
>   ([HuggingFace commits](https://huggingface.co/Salesforce/GTA1-7B/commits/main)).
>
> Since all of these results became available only after the ICLR deadline, they are considered concurrent work and were not available to us at submission time; thus they were not included in the comparisons in the main paper.
>
> As for your suggestion that:
>
> > it would strengthen the work to demonstrate whether the proposed iterative tool-coordination approach can further improve these stronger base models
>
> we would like to clarify that we did experiment with the first version of GTA1-7B as a base model. However, this initial GTA1-7B release had already been trained with RL on a $1.56$M-sample dataset, and at that time there were no other methods that used GTA1-7B as a starting point. This made it difficult to cleanly isolate and compare the effect of our approach against other training procedures on the same base model.
> Since you explicitly requested these results, we now report them below:
>
> | Model                               | Accuracy on ScreenSpot-Pro (%) |
> |:-----------------------------------:|:------------------------------:|
> | GTA1-7B (first released version)    | $50.1$                           |
> | GUI-Spotlight (initialized from GTA1-7B) | $55.7$                           |
>
>
> For ***Weaknesses 2:***
>
> > Data Efficiency Claims
>
> Thank you for raising this point. First, we would like to clarify that many models in **Table 3**
>  are also trained on top of UI-TARS-1.5-7B. For example, GTA1-7B starts from UI-TARS-1.5-7B and applies RL on $1.56$M interaction samples, yet achieves $50.1$% on ScreenSpot-Pro, which is still lower than our $52.8$%. In addition, the original UI-TARS-1.5-7B checkpoint attains $38.7$% on ScreenSpot-Pro, and our method improves this to $52.8$% using only $18.5$K additional training samples.
>
> These results provide strong evidence that our proposed procedure is effective and data-efficient, even when starting from an already strong GUI-pretrained base model.
>
> For ***Weaknesses 3 & Questions 2:***
>
> > For example, UI-AGILE achieves 48.7% on ScreenSpot-Pro with only 9K samples using Qwen2.5-VL as the base
>
> We are grateful that the first author of UI-AGILE kindly left comments in the Public Comments channel, and we have engaged in a discussion under their thread about the respective strengths and limitations of GUI-Spotlight and UI-AGILE.
>
> Here we briefly summarize this for the reviewers.
> UI-AGILE can be broadly characterized as a coarse prediction → rule-based cropping → selection pipeline. In **Section 5.4** of our paper, we also compare against this style of visual localization using the same base model. The results show that, when starting from UI-TARS-1.5-7B, this UI-AGILE-style method indeed improves performance, but it can also introduce issues such as cropping away parts of the original image that contain crucial information. GUI-Spotlight is explicitly designed to address this limitation by performing iterative focusing without permanently discarding context.
>
> In addition, **reviewer yySw** raised a similar question. In our response to that reviewer, we included a schematic figure that explains this issue in more detail; you may refer to the **second message in that discussion** for a visual illustration.
>
> Regarding your question of why our method only reaches $38.7$% when initialized from Qwen2.5-VL-7B, but improves to $52.8$% when initialized from UI-TARS-1.5-7B, we carefully analyzed the error cases. For Qwen2.5-VL-7B, most failures stem from unsuccessful tool invocations, whereas this issue is largely absent for UI-TARS-1.5-7B. We attribute this gap to the fact that UI-TARS-1.5-7B is further pre-trained on substantial agentic / tool-use–style data on top of Qwen2.5-VL-7B, which is the current natural trend in LLM pre-training and allows it to quickly acquire and generalize to new GUI tools under our training procedure.

---

> > ### Author Response · Authors · 2025-11-25
> >
> > For ***Weaknesses 4 & Questions 4:***
> >
> > > Have the authors considered extending the tool set to leverage GUI-specific structure
> >
> > Thank you for your suggestions on improving our tool set. In fact, when designing the tools we did experiment with various page-parsing and OCR-style utilities, including EasyOCR and OmniParser. However, we ultimately decided not to include these tools in our final tool set for the following reasons:
> >
> > 1. **Limited benefit of explicit OCR tools.**
> >    For OCR tools, we found that whenever the OCR system can reliably extract the target text from a region, the underlying vision–language model can typically read that text directly from the image **without** an additional tool call. As a result, the marginal benefit of adding a separate OCR tool was limited.
> >
> > 2. **Unreliable structure-aware parsers on real GUIs.**
> >    For more structure-aware parsers such as OmniParser, we observed that on many real-world, highly stylized interfaces they sometimes return incorrect or noisy element predictions. Such erroneous signals can mislead the agent’s reasoning instead of helping it, which is undesirable in our RL setting.
> >
> > For these reasons, we chose a **minimal set of low-level but reliable tools**—`crop`, `extract`, and `find_color`—whose primary role is to provide precise local visual context. We then rely on training to let the model itself learn to interpret and reason about the various UI elements within these regions.
> >
> > For ***Questions 3:***
> >
> > > What is the distribution of tool invocations during inference?
> >
> > Thank you for raising this question. We have computed detailed statistics of tool usage on both the ScreenSpot-Pro and UI-Vision benchmarks, as summarized in the tables below.
> >
> > **Tool-call statistics across benchmarks**
> >
> > | Benchmark      | # Cases | Cases with 1 tool call | Cases with 2 tool calls | Avg. tool calls / case |
> > |:--------------:|:-------:|:----------------------:|:-----------------------:|:----------------------:|
> > | ScreenSpot-Pro |  $1581$   |         $1498$           |           $83$            |         $1.05$           |
> > | UI-Vision      |  $1460$   |         $1407$           |           $53$            |         $1.04$           |
> >
> > **Per-tool usage**
> >
> > | Benchmark      | `extract` calls | `crop` calls | `find_color` calls | Total tool calls |
> > |:--------------:|:---------------:|:------------:|:------------------:|:----------------:|
> > | ScreenSpot-Pro |      $1523$       |     $131$      |         $10$         |       $1664$       |
> > | UI-Vision      |      $1378$       |     $132$      |          $3$         |       $1513$       |
> >
> > For ***Weaknesses 5 & Questions 5:***
> >
> > > Fixing the presentation issues noted in the weaknesses would improve clarity and readability.
> >
> > Thank you for this suggestion. In the revised version, we have updated several formatting details in line with your comments, including using the appropriate `\citep` command for parenthetical citations, consistently writing `GUI-Spotlight` instead of `\textsc{GUI-Spotlight}` (except in headings), and adjusting the spacing between figures/tables and the surrounding text to improve readability.
> >
> > ----
> >
> > Once again, thank you for all of your thoughtful questions, and we look forward to your further feedback.

---

### Official Review · Reviewer_jH6a · 2025-10-30

**Soundness:** 3
**Presentation:** 3
**Contribution:** 2
**Rating:** 6
**Confidence:** 3

**Summary:**

Introduces GUI-Spotlight:
1. A dataset pipelines that filters be high quality using Qwen2.5 (removing low quality instructions, inaccurate bounding boxes and consistency).
    a. This includes introducing a new dataset of 15k high-resolution samples, since existing open-source datasets did not have high resolution images.
2. Introduces a multi-stage fine-tuning setup with SFT and RL. Ablating various RL setups.
3. Outperforms existing models on GUI-specific benchmarks, using few samples.

**Strengths:**

1. Introduces an interesting new training setup for tool-use in GUI-setups. Ablates various RL algorithms, and shows how they are stable or  unstable.
2. Various ablations on their training steps.
3. Good performance beyond the evaluation set they hillclimbed on.

**Weaknesses:**

1. Details are unclear about their new proposed dataset:
    a. 15k high resolution samples, but where did they come from? How were the domains selected?
    b. Where did the instructions come from in this crawled dataset?
    c. "For our high-resolution dataset, we enhanced the pipeline with additional functions" -> details?
2. The new dataset sounds very useful, but it wasn't ablated how much of the performance improvement is simply because of the new data pipeline or high quality samples that were crawled outside of the open-source datasets. Hard to estimate the impact of that versus the other proposals (like multi-stage or RL algorithm).

**Questions:**

1. Can you add more details on the dataset filtering and dataset creation from scratch (e.g. in appendix if no space)?
2. Can you ablate training only on the newly filtered + proposed dataset, ignoring tool calls or multi-stage finetuning?
3. How much does this fine-tuned model decrease in performance on non-GUI tasks? Can we still use it as a "foundation model"? Even if performance in non-gui tasks is bad, would be helpful to report (none of the existing GUI-specific frameworks keep generalization beyond GUI, but it's not being tracked).
4. "After warm-up the model learns to invoke multiple tools but remains under-aligned": Do we know whether we trained for long enough/hyper-parameters are optimal (or "good-enough")? I don't see many experiments around stage 1 (completely removing it, increasing duration, different HPs, different data)
5. How much compute did you use?

---

> ### Author Response · Authors · 2025-11-25
>
> We thank the reviewer for the thoughtful and constructive feedback. Below, we provide point‑by‑point responses to the comments.
>
> ----
>
> For ***Weaknesses 1 & Questions 1:***
>
> > Details are unclear about their new proposed dataset
>
> Thank you for the question. In response to points:
> > (a) “15k high resolution samples, but where did they come from? How were the domains selected?”
> > (b) “Where did the instructions come from in this crawled dataset?”,
>
> We have added a dedicated **Appendix A.4** describing the construction of our high-resolution dataset in detail, including the stages of data sources and domain selection, element discovery and filtering, bounding box extraction and alignment to the screenshot, and instruction construction.
>
> For point
> > (c) “For our high-resolution dataset, we enhanced the pipeline with additional functions”
>
> Our data-cleaning procedure largely follows the steps in **Section 3.2.1** (Dataset cleaning part). The main difference is that, for the high-resolution samples crawled directly from the web, we additionally filter out images from inappropriate or unsafe domains.
>
> In addition, as described in **Section 3.2.2**, our training procedure consists of three stages. The first two stages use only our filtered UGround dataset, and after these two stages the model already reaches $49.6$% accuracy on ScreenSpot-Pro. In the final stage, we further fine-tune the model using only $4$K samples randomly selected from the $15$K high-resolution dataset. This three-stage schedule and the corresponding performance gains are summarized in **Figure 2**.
>
> For ***Weaknesses 2 & Questions 2:***
>
> > The new dataset sounds very useful, but it wasn't ablated how much of the performance improvement is simply because of the new data pipeline or high quality samples that were crawled outside of the open-source datasets.
>
> Thank you for the question. We would like to emphasize that, based on our experiments, the performance gains of GUI-Spotlight are primarily attributable to the algorithmic improvements rather than merely to higher-quality data.
>
> As described in **Section 3.2.2**, the first two training stages use only data filtered from the UGround dataset. The *high quality samples that were crawled outside of the open-source datasets* mentioned in your Weakness section correspond to the $4$K high-resolution samples used only in the third stage. As shown in **Figure 2**, after completing the second stage, the model’s accuracy on ScreenSpot-Pro already increases from $17.5$% to $49.6$%. In the third stage, we further fine-tune on $4$K high-resolution cases randomly sampled from the $15$K pool; across multiple random seeds, the final performance consistently falls within $52.3$%–$53.6$%. The goal of adding these high-resolution cases is mainly to encourage continued exploration during RL training and to prevent the policy from converging prematurely, rather than to rely on them as the dominant source of improvement.
>
> Moreover, **Figure 3** directly compares different reinforcement learning algorithms under the same training data. As shown in the right panel of **Figure 3**, even with identical datasets, the original GRPO and GSPO baselines start to degrade after around $275$ training steps, whereas our proposed method continues to improve. This indicates that the accuracy gains of GUI-Spotlight are largely driven by the proposed training algorithm, rather than by the introduction of new data alone.
>
>
> In addition, as requested, we also trained a model without tool calls and without multi-stage finetuning. The results are summarized below:
>
>
> | Model                               | ScreenSpot-Pro Acc. (\%) |
> |:-------------------------------------:|:--------------------------:|
> | UI-TARS-1.5-7B (original baseline)  |  $38.7$                    |
> | Ours (no tools, single-stage fine-tune) |$41.2$                  |
>
> This result clearly shows that simply fine-tuning the model on a small $18.5$K dataset, without tools or multi-stage training, only yields a modest improvement in visual grounding accuracy. In contrast, the main performance gains of GUI-Spotlight come from the proposed algorithmic design, rather than from the data alone.

---

> > ### Author Response · Authors · 2025-11-25
> >
> > For ***Weaknesses 3:***
> >
> > Thank you for raising this concern about non-GUI capabilities. To quantify the impact of our GUI-specific fine-tuning, we evaluated both the base Qwen2.5-VL-7B model and our GUI-Spotlight checkpoint (fine-tuned from Qwen2.5-VL-7B) on the MMLU benchmark. The results are summarized below:
> >
> > | Model                                      | MMLU Accuracy (\%) |
> > |:------------------------------------------:|:------------------:|
> > | Qwen2.5-VL-7B (base)                       | $68.0$               |
> > | GUI-Spotlight (fine-tuned from Qwen2.5-VL-7B) | $64.0$               |
> >
> > Thus, fine-tuning on GUI grounding leads to a modest absolute drop of $4.0$ points on MMLU, while the model still retains reasonably strong general-domain performance.
> >
> > For ***Weaknesses 4:***
> >
> > > Do we know whether we trained for long enough/hyper-parameters are optimal (or "good-enough")?
> >
> > Thank you for pointing this out. In fact, we experimented with three different SFT data scales for the warm-up stage: $2561$, $8$K, and $15$K samples. In the paper, we only reported the results for the $2561$-sample warm-up, which achieved the best performance. We observed that, for this task, using more SFT data led to **lower final accuracy after RL**, with training quickly plateauing at a suboptimal level.
> >
> > We hypothesize that this is because a large amount of SFT over-constrains the policy: the model becomes overly confident around the supervised data distribution (low-entropy, narrow modes). Given a limited RL budget (steps, learning rate, and exploration), it then becomes difficult for the model to escape this initialization and discover better strategies, causing it to get stuck in a local optimum.
> >
> > The detailed experimental results are as follows.
> >
> > | # SFT samples (Stage 1) | Final accuracy after RL (%) | Observed behavior |
> > |:-----------------------:|:---------------------------:|:------------------------------------------:|
> > | $2,561$                   | $52.8$                        | Best performance; good balance & generality |
> > | $8,000$                   | $48.5$                        | Converges to a lower plateau               |
> > | $15,000$                  | $45.2$                        | Strong plateau at low accuracy, loses generality |
> >
> > For ***Weaknesses 5:***
> >
> > > How much compute did you use?
> >
> > The detailed hardware configuration is provided in the first part of **Appendix A.3**. Each full training run uses $4$ NVIDIA H200 GPUs for approximately $36$ hours, mainly due to the large number of multi-turn interactions and tool calls involved.
> >
> > ----
> >
> > Once again, thank you for all of your thoughtful questions, and we look forward to your further feedback.

---

### Official Review · Reviewer_yySw · 2025-11-01

**Soundness:** 3
**Presentation:** 3
**Contribution:** 3
**Rating:** 4
**Confidence:** 5

**Summary:**

GUI-SPOTLIGHT proposes an iterative "spotlight" mechanism for GUI visual grounding, where a multimodal model invokes cropping, quadrant extraction, and color-guided tools to progressively refine focus on screen regions. It couples multi-turn tool reasoning with a stabilized GSPO-based reinforcement learning pipeline and curated high-resolution GUI data. The approach improves grounding accuracy on ScreenSpot-Pro and UI-Vision while using far fewer samples than prior systems. Contributions include the iterative tool framework, modified multi-tool RL objective, and dataset/ablation insights for GUI grounding.

**Strengths:**

1. The paper implements a structured, multi-turn GUI grounding model with crop, quadrant, and color-guided tools, paired with a stabilized GSPO-based RL objective, offering a disciplined training-and-tool framework for GUI localization.

2. Empirical results on ScreenSpot-Pro and UI-Vision, along with ablations on tool policies and reward shaping, show consistent accuracy gains and provide useful insights into scaling high-resolution GUI grounding with limited data.

**Weaknesses:**

1. The core "spotlight" idea and iterative tool chain (crop/quadrant/color) closely follow prior ScreenSpot-Pro iterative narrowing and ScreenSeekeR baselines, and the main novelty lies largely in data curation and retraining rather than a fundamentally new paradigm. A clearer comparison and attribution to SS-Pro’s iterative region-refinement design is needed.

2. Directly comparing scores shows GUI-Spotlight performs far below GTA-1-7B when using the same Qwen2.5-VL-7B backbone; without closing that gap or analyzing failure modes, the method provides limited evidence it advances the state of the art. A detailed head-to-head and explanation of the performance deficit are necessary.

3. The paper evaluates only isolated grounding tasks and lacks validation in the GUI agents tasks (e.g., OSWorld). Demonstrating consistent performance in holistic GUI tasks and multi-step interaction settings is necessary to support claims of practical utility.

**Questions:**

Apart from the training process, what differences exist between this work and the ScreenSeekeR baseline in ScreenSpot-Pro?

---

> ### Author Response · Authors · 2025-11-23
>
> We thank the reviewer for the thoughtful and constructive feedback. Below, we provide point‑by‑point responses to the comments.
>
> ----
>
> For ***Weaknesses 1 & Questions 1:***
>
> > The core "spotlight" idea and iterative tool chain (crop/quadrant/color) closely follow prior ScreenSpot-Pro iterative narrowing and ScreenSeekeR baselines
>
> Thank you for the question. However, we would like to emphasize that GUI-Spotlight is fundamentally different from the common *coarse prediction → fixed-rule cropping → fine prediction* visual grounding paradigm, such as the ScreenSeekeR pipeline you mentioned. **In fact, GUI-Spotlight is explicitly designed to address the potential failure modes inherent in this type of fixed procedural cropping approach.**
>
> In **Section 5.4** of our paper, we explicitly compare GUI-Spotlight with ScreenSeekeR-style visual grounding. Concretely, we evaluate three settings on ScreenSpot-Pro:
>
> 1. Procedural (single-view history, no training).
>
> A variant analogous to ScreenSeekeR: we first obtain a coarse prediction on the full image, then crop a fixed-size region around that prediction, and finally run a refined prediction on the cropped image. At each step, only the current cropped image is included in the model’s context, and the entire procedure is purely inference (no additional training). This corresponds to **Fig. 5 ②** in our paper.
>
> 2. Procedural (multi-view history, no training).
>
> The same rule-based coarse-then-crop procedure as above, but every intermediate image (the original screenshot and all cropped patches) is appended to the dialogue history, similar to how GUI-Spotlight accumulates observations. This is again a purely inference variant without any additional training, corresponding to **Fig. 5 ①**.
>
> 3. GUI-Spotlight (ours).
>
> Cropping tools (extract, find_color, crop) are exposed as explicit actions. The agent learns, via RL, when and how to invoke these tools while maintaining access to both global and local visual context (**Fig. 5 ③**).
>
> Our experiments show that a purely procedural scheme similar to ScreenSeekeR does improve accuracy by about 10 percentage points under our setting. However, it also causes failures on some cases that the base model could solve in a zero-shot manner. **Inspecting these cases, we find that when the crop is determined purely by fixed rules and only the cropped image is fed to the model, the model can sometimes lose crucial global context, which leads to incorrect predictions despite having a roughly correct coarse localization.**

---

> > ### Author Response · Authors · 2025-11-23
> >
> > For example: Here is an example compare ScreenSeekeR-style visual grounding vs. GUI-Spotlight
> >
> > **ScreenSeekeR-style visual grounding:**
> >
> > This is the Original high-resolution screen (with a long search bar):
> >
> > ```text
> > +-------------------------------------------------------------+
> > |  ...                                                        |
> > |                                                             |
> > |        [=================  Search Bar  =================]   |
> > |                                                             |
> > |  ...                                                        |
> > +-------------------------------------------------------------+
> > ```
> >
> > ScreenSeekeR-style: fixed-rule cropping → only a small fragment is shown to the model:
> >
> > ```text
> >                        Fixed-window crop
> >                 +-------------------------+
> >                 | ====  partial bar  ==== |   ← just a partial segment of the bar
> >                 +-------------------------+
> > ```
> >
> > → The model only sees this small fragment, so it is hard to tell:
> >   - Is this part of a search bar?
> >   - A button?
> >   - A progress bar / slider?
> >
> >
> > **GUI-Spotlight**
> >
> > Step 1: First look at the full image and decide to use which tool
> >
> > ```text
> > +-------------------------------------------------------------+
> > |  ...                                                        |
> > |        [=================  Search Bar  =================]   |
> > |                  ↑ agent autonomously chooses crop region   |
> > +-------------------------------------------------------------+
> > ```
> >
> > Instead of using a fixed cropping pattern, the model autonomously chooses where to crop, allowing it to correctly capture the entire long bar
> >
> > ```text
> >                        autonomously chosen crop region
> >                 +----------------------------------------------------+
> >                 | [=================  Search Bar  =================] |   ← entire long bar
> >                 +----------------------------------------------------+
> >
> > ```
> >
> > Step 2: Call tool to obtain a local view, while keeping the global view
> >         in the dialogue history
> >
> >
> > The global image remains in the context:
> >
> > ```text
> > +-------------------------------------------------------------+
> > |  ...                                                        |
> > |        [=================  Search Bar  =================]   |
> > |  ...                                                        |
> > +-------------------------------------------------------------+
> > ```
> >
> > The cropped image is appended as a new observation:
> >
> > ```text
> > +----------------------------------------------------+
> > | [=================  Search Bar  =================] |
> > +----------------------------------------------------+
> > ```
> >
> > → During reasoning, the model can refer to:
> >   - The global image: to understand that this is part of an entire search bar;
> >   - The local image: to see local details for precise clicking.
> >
> > → RL learns *how much to crop / when to stop cropping*, rather than being constrained
> >   by a single fixed crop window size.
> >
> >
> > A natural question is whether we could simply adopt a ScreenSeekeR-style inference procedure while also add the full global image to the top of the dialogue history. In our setting, however, doing so without any additional training not only fails to improve accuracy, but in fact dramatically degrades it: as shown in **Fig. 5** of our paper, this variant causes the accuracy on ScreenSpot-Pro to drop sharply to just 7.6%.
> >
> >
> > In summary, GUI-Spotlight treats cropping as an explicit tool that can be applied to arbitrarily chosen regions of the original screenshot, rather than, as in ScreenSeekeR, only feeding the model locally cropped patches produced by a fixed cropping pattern at each step. This design allows the model not only to remove visual noise through cropping, but also to retain global context in the dialogue history and autonomously decide where and how much to crop, which in turn leads to higher overall visual grounding accuracy.

---

> ### Author Response · Authors · 2025-11-23
>
> For ***Weaknesses 2:***
>
> > Directly comparing scores shows GUI-Spotlight performs far below GTA-1-7B when using the same Qwen2.5-VL-7B backbone
>
> Thank you for raising this point. We would like to clarify that GTA1-7B in fact uses the same base model as ours, **UI-TARS-1.5-7B**, rather than a vanilla Qwen2.5-VL-7B backbone as stated in Weaknesses 2.
>
> In the GTA1 paper (https://arxiv.org/abs/2507.05791), they said:
>
> > Our model is initialized from UI-TARS-1.5-7B, Qwen2.5-VL-32B-Instruct, and Qwen2.5-VL-72B-Instruct.
>
> In addition, it is worth noting that GUI-Spotlight is trained on only **18.5K** data samples, whereas GTA1 is trained on **1.56M** samples. This highlights that GUI-Spotlight achieves strong data efficiency, reaching competitive performance while using nearly two orders of magnitude fewer examples.
>
>
> For ***Weaknesses 3:***
>
> > The paper evaluates only isolated grounding tasks and lacks validation in the GUI agents tasks (e.g., OSWorld).
>
> Thank you for the question. Although GUI-Spotlight is primarily focused on the core capability of visual grounding rather than on building and evaluating a full end-to-end GUI agent, we also evaluate its effectiveness on real-world GUI tasks.
>
> We first follow the [Qwen3-VL](https://camo.githubusercontent.com/7b30ae6af5402a71bc63076de6e1b67d5bf94c1ada1c3fb59fddaf30b98fbd8a/68747470733a2f2f7169616e77656e2d7265732e6f73732d616363656c65726174652e616c6979756e63732e636f6d2f5177656e332d564c2f7461626c655f6e6f7468696e6b696e675f766c2e6a7067)
>  evaluation setup and measure GUI-Spotlight’s performance on OSWorld-G, as reported in **Section 5.3**.
>
> In addition, we conducted a agent evaluation using the [InfantAgent](https://arxiv.org/abs/2505.10887) framework, with Claude 4.5 sonnet as the planner and either **GUI-Spotlight** or **UI-TARS-1.5-7B** as the GUI grounding module. Over 50 steps, replacing the original UI-TARS-1.5-7B grounding with GUI-Spotlight yields a modest but consistent improvement in task success rate:
>
> | Grounding model            | Agent success rate on OSWorld (%) |
> |:--------------------------:|:----------------------------------:|
> | UI-TARS-1.5-7B (baseline)  |              $58.17$                 |
> | GUI-Spotlight              |              $60.90$                 |
>
>
> ----
>
> Once again, thank you for all of your thoughtful questions, and we look forward to your further feedback.

---

### Public Comment · ~Shuquan_Lian2 · 2025-11-13

Dear Authors,

I enjoyed reading your paper. The idea of formulating "Crop" as an active tool invoked by the model is a very compelling design choice.

I am writing to suggest a comparison with a relevant recent work, UI-AGILE: Advancing GUI Agents with Effective Reinforcement Learning and Precise Inference-Time Grounding, which also explores the importance of cropping for GUI grounding (evaluated on the shared ScreenSpot-Pro benchmark).

Unlike your approach where the agent actively decides when to crop, UI-AGILE implements cropping as a procedural mechanism (implicitly integrated into training and inferencing, not invoked by the model) to reduce the visual noise.

Discussing these two distinct paradigms—active tool usage vs. procedural processing—would provide readers with a broader perspective on how to handle high-resolution GUI inputs effectively.

Best regards

---

> ### Author Response · Authors · 2025-11-22
>
> Hello Shuquan!
>
> Thank you very much for your comment, and my apologies for the late reply. We have carefully read your UI-AGILE paper, and please let us know if the following understanding is inaccurate.
>
> As we understand it, cropping in UI-AGILE is implemented as a fixed procedural component outside the model’s action space. During training, when all sampled predictions for a given example receive zero grounding reward, a cropping-based resampling strategy is applied: using the ground-truth bounding box, the system scans the screenshot with a window of pre-defined size and stride to obtain a cropped image that fully contains the target, and then resamples predictions on this easier view.
>
> At inference time, the pipeline first performs a coarse prediction on sub-images obtained by a fixed decomposition of the full screenshot. Around each predicted point, the system then crops an element image using hand-crafted rules (e.g., a fixed-size box centered at the point). A separate Yes/No classifier is used to score these element images against the instruction, and the candidate with the highest score is selected as the final prediction. In other words, cropping and refinement are realized as a procedural “coarse prediction → rule-based cropping → selection” process, rather than as explicit actions chosen by the agent.
>
> In Section 5.4 of our paper, we compare GUI-Spotlight with this type of coarse-then-refine procedural processing, though our comparison only implements inference-time variants and does not include your reinforcement fine-tuning stage. Concretely, we evaluate three settings on ScreenSpot-Pro:
>
> 1. Procedural (single-view history, no training)
>
> A variant analogous to the pipeline you described: we first obtain a coarse prediction on the full image, then crop a fixed-size region around that prediction, and finally run a refined prediction on the cropped image. Only the currently used image (either the original or the cropped one) is included in the model’s context at each step, and this procedure is purely inference-time (no additional training). This corresponds to Fig. 5② in our paper.
>
> 2. Procedural (multi-view history, no training)
>
> The same rule-based coarse-then-crop procedure as in (1), but every intermediate image (the original screenshot and the cropped images) is appended to the dialogue history, similar to how GUI-Spotlight accumulates observations. Again, this is a purely inference-time variant without training. This corresponds to Fig. 5①.
>
> 3. GUI-Spotlight (ours)
>
> Cropping tools (extract, find_color, crop) are exposed as explicit actions. The agent learns, via RL, when and how to invoke these tools while maintaining access to both global and local visual context (Fig. 5③).
>
> Our experiments confirm your insight that such procedural cropping can effectively denoise high-resolution GUI screenshots: the first procedural variant improves accuracy on ScreenSpot-Pro by roughly 10 percentage points over the base model without cropping.
>
> However, we also analyzed the failure cases of this strategy. When the crop is determined purely by fixed rules and only the cropped image is fed to the model, the model can sometimes lose crucial global context. For example, when localizing a long search bar, a fixed crop may capture only a fragment of the bar, making it ambiguous what the cropped region actually corresponds to in the overall interface.
>
> This observation is one of the motivations behind GUI-Spotlight: we aim to let the agent decide when and where to crop while still retaining access to the global view, so that it can denoise the visual input in a more adaptive, task-aware manner and ultimately perform more accurate grounding.
>
> Thank you again for pointing us to your work. We will cite UI-AGILE and clarify this connection in Section 5.4. If we have misunderstood any part of your method, we would be very grateful for your corrections. We would also be happy to discuss any further ideas or potential extensions with you!

---

### Author Response · Authors · 2025-11-26

Dear Reviewers,

Thank you again for the time and effort you have devoted to reviewing our work. We have provided point-by-point responses to all the comments and questions raised, and have made corresponding revisions and clarifications in the manuscript. We look forward to further discussion. Thank you again.

---

> ### Author Response · Authors · 2025-11-27
>
> Dear reviewers:
>
> As the rebuttal deadline is approaching, we would really appreciate your feedback on our rebuttal so that we can iterate and improve the final version.
>
> We have also made a minor round of revisions to the manuscript. The main changes are as follows:
>
> 1. To provide a comprehensive view of recent progress in GUI grounding, we summarize complementary lines of work, including scaling, modularization, and training-free grounding. Accordingly, we have added citations to the following papers:
>
> > [1] Aria-ui: Visual grounding for gui instructions
>
> > [2] Improving fine-grained visual recognition in low data regimes via self-boosting attention mechanism
>
> > [3] Attention-driven gui grounding: Leveraging pretrained multimodal large language models without fine-tuning
>
> > [4] Semi-supervised semantic segmentation with prototype-based consistency regularization
>
> 2. Additionally, to facilitate reproducibility for other researchers, we clarify in the Reproducibility Statement that our training data are filtered from the UGround dataset, as described in the paper, and we thank the UGround team for open-sourcing the dataset.
>
> If you have time, please take a look at the rebuttal and share any comments. We thank the reviewers and AC again for their time!

---

### Author Response · Authors · 2025-11-30

We again thank the Area Chair and the reviewers for their time and effort in evaluating our submission. To help the AC quickly grasp the main contents of our rebuttal, we provide a concise summary of the reviewers’ questions and our corresponding responses below:

---

***Summary of responses to Reviewer yySw***

1. Weakness 1 & Question 1

**Reviewer question:** The reviewer questioned how our approach differs from ScreenSeekeR-style baselines.

**Our response:** We clarified the key conceptual and practical differences between GUI-Spotlight and fixed-rule, procedural cropping pipelines, and we emphasized that we already included controlled comparisons against ScreenSeekeR-style baselines in **Section 5.4**.

2. Weakness 2

**Reviewer question:** GUI-Spotlight performs below GTA-1-7B when using the same Qwen2.5-VL-7B backbone.

**Our response:** We corrected this factual inaccuracy: **GTA-1-7B is trained with UI-TARS-1.5-7B** as its initialization backbones, i.e., it is not initialized from a plain Qwen2.5-VL-7B setup. We also highlighted that our method achieves strong performance under the same base-model family while being substantially more data-efficient.

3. Weakness 3

**Reviewer question:** The paper evaluates only isolated grounding tasks and lacks validation on GUI agent tasks (e.g., OSWorld).

**Our response:** While building a full autonomous GUI agent is beyond this paper’s scope (our focus is training a think-with-image grounding model), we nonetheless added OSWorld results as requested to demonstrate downstream utility in agent-style settings.

***Summary of responses to Reviewer jH6a***

1. Weakness 1 & Question 1

**Reviewer question:** The reviewer requested more details on how the proposed dataset was created.

**Our response:** We added a detailed description of the dataset construction pipeline in **Appendix A.4**.

2. Weakness 2 & Question 2

**Reviewer question:** The reviewer suggested our improvements might be primarily driven by the dataset.

**Our response:** We pointed to evidence in the paper showing that the performance gains are attributable to the algorithmic design, not merely the data.

3. Weakness 3

**Reviewer question:** The reviewer asked about the model’s performance on general (non-GUI) tasks.

**Our response:** We added MMLU comparisons as requested.

4. Question 4

**Reviewer question:** The reviewer asked how varying the SFT warm-up data size affects performance.

**Our response:** We explained that the main paper reports results with the 2561-sample setting due to space constraints, and we additionally reported the final performance trends across different warm-up data scales.

5. Question 5

**Reviewer question:** The reviewer asked about the computational resources we used.

**Our response:** Beyond the hardware information already reported in **Appendix A.3**, we added further implementation/compute details to facilitate reproducibility.

---

> ### Author Response · Authors · 2025-11-30
>
> ***Summary of responses to Reviewer Kbre***
>
> 1. Weakness 1 & Question 1
>
> **Reviewer question:** The reviewer noted that we did not compare our work with several recent works.
>
> **Our response:** We clarified that the works listed by the reviewer were released after the ICLR submission deadline, and thus could not be included in the original comparison at submission time.
>
> 2. Weakness 2
>
> **Reviewer question:** The reviewer challenged our data-efficiency claims because we train starting from UI-TARS-1.5-7B.
>
> **Our response:** We reiterated that **Table 3** already compares against multiple methods also built on UI-TARS-1.5-7B, supporting our data-efficiency conclusions under a fair base-model setting.
>
> 3. Weakness 3 & Question 2
>
> **Reviewer question:** The reviewer asked about the differences between our work and UI-AGILE.
>
> **Our response:** We explained distinctions between our approach and UI-AGILE using illustrative diagrams, quantitative comparisons, and design-level discussion, as requested.
>
> 4. Weakness 4 & Question 4
>
> **Reviewer question:** The reviewer asked why we did not include additional tools such as OCR or structure-aware parsers in our toolset.
>
> **Our response:** We noted that we experimented with such tools, and we explained why they were not adopted in the final toolset (based on effectiveness and design considerations).
>
> 5. Question 3
>
> **Reviewer question:** The reviewer requested the distribution of tool invocations during inference.
>
> **Our response:** We added the requested statistics on tool-invocation distributions during inference.
>
> 6. Weakness 5 & Question 5
>
> **Reviewer question:** The reviewer raised several presentation/clarity issues.
>
> **Our response:** We addressed these comments and made corresponding edits to improve clarity and correctness.
>
> ***Summary of responses to Reviewer 5C7T***
>
> 1. Weakness 1
>
> **Reviewer question:** The reviewer asked for a more detailed explanation of how the model invokes tools.
>
> **Our response:** We expanded the explanation of how the model invokes tools and clarified the role and behavior of each tool.
>
> 2. Weakness 2
>
> **Reviewer question:** The reviewer requested details on how the SFT warm-up data were obtained.
>
> **Our response:** We added further details to Section 3.2.2 describing how the SFT warm-up data were collected/constructed.
>
> 3. Weakness 3
>
> **Reviewer question:** The reviewer requested the distribution of tool invocations during inference.
>
> **Our response:** We provided the requested tool-invocation distribution statistics.
>
> 4. Weakness 4
>
> **Reviewer question:** The reviewer asked whether the extract/crop tool upsamples (enlarges) the cropped region.
>
> **Our response:** We clarified that extract/crop tools do **NOT** upsample/enlarge the cropped region. Our main motivation is not to compensate for resolution compression, but to remove irrelevant regions and focus the model on the area around the target element.
>
> 5. Weakness 5
>
> **Reviewer question:** The reviewer questioned why our model underperforms GUI-Actor.
>
> **Our response:** We explained that our training uses approximately $1/500$ of the data compared to GUI-Actor, which largely accounts for the performance gap, and we highlighted our method’s data-efficiency perspective.
>
> 6. Weakness 6
>
> **Reviewer question:** The reviewer stated that the paper lacks exploration of GUI agent systems.
>
> **Our response:** Although full GUI agent-system development is beyond the scope of this paper, we nevertheless added OSWorld results to demonstrate relevance to agent-style downstream tasks.

---

### Meta-Review · Area_Chair_GReL · 2026-01-07

**Summary:**

The main concerns from the reviewers are following:

- Reviewer **yySw**:
  - **W1**: The core idea and paradigm share similarity with previous approaches.
  - **W2**: Worse performance against GTA-1-7B.
  - **W3**: Lacking validation in GUI agent tasks.

- Reviewer **jH6a**:
  - **W1**: For the performance gain, how the credit is assigned between the collected dataset and the proposed training algorithm.

- Reviewer **Kbre**:
  - **W1**: The data efficiency builds upon relatively strong base models.
  - **W2**: Worse performance against UI-AGILE.
  - **W3**: Relatively limited tool design.

- Reviewer **5C7T**:
  - **W1**: Concerns on tool functions.
  - **W2**: The computational overhead from element grounding tool calls, especially when compared with existing baselines.
  - **W3**: Context loss for localizing operations.
  - **W4**: Lacking evaluation on GUI agent tasks.

**Reviewer Concerns:**

- **Concerns addressed in the rebuttal:**
  - Reviewer **yySw**: The authors point out the factual error regarding the base model of GTA-1-7B in **W2**.
  - Reviewer **jH6a**: The authors provide detailed explanations on the effectiveness of the algorithm beyond collected data.
  - Reviewer **Kbre**: The authors respond detailedly to address the concerns regarding base models and tool design in **W1** and **W3**.
  - Reviewer **5C7T**: The authors respond detailedly to address the concerns regarding tool function design (**W1**) and potential context loss (**W3**).

- **Concerns remained outstanding:**
  - Reviewers raise their concerns about the comparison of existing works from different points, such as the similarity of paradigm, failure of beating the SOTA, and the computational overhead. The author responses provide detailed discussions to clarify the difference and strength of the proposed method, while I think the concerns are not fully eliminated: 1) The authors argue that a major advantage is that even though both the proposed method and existing approaches utilize localize and crop operations, they are learned instead of pre-defined, which is somehow a solid engineering improvement rather than a strong research advancement; 2) The issue of computational overhead is not fully addressed by the additional experimental results provided in the response to **W2** of Reviewer **5C7T**. The results show that the learned model usually conducts limited number of tool use operations, mostly only once per task. Even though this shows that the computational overhead resulted from tool use is relatively low, while this also in contradiction to conducting **iterative focus refinement**, which is claimed as a major contribution. In my view, these two issues require more in-depth justification.

  - Reviewers also raise concern about the lacking evaluation on GUI agent tasks. The authors argue that this is beyond the focus of the paper, which I agree if the technical contribution of the paper is soundly justified. While due to the current remaining concerns as stated in the first point, I also treat this as a secondary weak point.

**Reviewer Scores:**

As discussed in the Reviewer Concerns, some points are indeed addressed by the author responses. While in my view, the remaining concerns (**W1** for Reviewer **yySw**, **W2** for Reviewer **5C7T**) also prevent the reviewers with negative evaluations to fully change their opinions.

---

### Decision · Program_Chairs · 2026-01-26

Reject